# Linking emotional valence and anxiety in a mouse insula-amygdala circuit

C. Nicolas [1], A. Ju [1], Y. Wu[1], H. Eldirdiri [1], S. Delcasso[1], Y. Couderc[1], C. Fornari[1], A. Mitra[1], L. Supiot [1], A. Vérité[1], M. Masson[1], S. Rodriguez-Rozada [2], D. Jacky[1], J. S. Wiegert [2] & A. Beyeler [1] ✉

Responses of the insular cortex (IC) and amygdala to stimuli of positive and negative valence are altered in patients with anxiety disorders. However, neural coding of both anxiety and valence by IC neurons remains unknown. Using fiber photometry recordings in mice, we uncover a selective increase of activity in IC projection neurons of the anterior (aIC), but not posterior (pIC) section, when animals are exploring anxiogenic spaces, and this activity is proportional to the level of anxiety of mice. Neurons in aIC also respond to stimuli of positive and negative valence, and the strength of response to strong negative stimuli is proportional to mice levels of anxiety. Using ex vivo electrophysiology, we characterized the IC connection to the basolateral amygdala (BLA), and employed projection-specific optogenetics to reveal anxiogenic properties of aIC-BLA neurons. Finally, we identified that aIC-BLA neurons are activated in anxiogenic spaces, as well as in response to aversive stimuli, and that both activities are positively correlated. Altogether, we identified a common neurobiological substrate linking negative valence with anxiety-related information and behaviors, which provides a starting point to understand how alterations of these neural populations contribute to psychiatric disorders.

Anxiety is defined as the anticipation of a future threat, with an uncertain probability of occurrence[1–3]. Importantly, anxiety is a physiological and adaptive state, evolutionarily relevant, since it allows organisms to prevent exposure to harmful situations. Anxiety becomes pathological when avoidance behaviors and fear are sustained and disruptive despite the absence of danger or potential danger[1,4]. Clinical studies demonstrated that patients with anxiety disorders have altered attribution of emotional valence, as they exhibited an attentional bias for stimuli of negative valence, as well as an increase in negative interpretations of ambiguous sentences and scenarios compared to healthy controls[5–7]. Consequently, it has been hypothesized for almost a decade that the neural circuits encoding anxiety and emotional valence overlap.

The insular cortex (IC, also named insula) has been shown to be involved in both valence processing and anxiety disorders[8] making this region a candidate structure that could encode both valence- and

anxiety-related behaviors. Indeed, a functional imaging study has revealed that, in healthy individuals, the insula exhibits opposing responses to stimulations of negative and positive valence, compared to neutral stimulations, with higher activity in response to aversive stimuli, and lower activity in response to rewarding stimulations[9]. Interestingly, this study, along with a meta-analysis of a dozen of other imaging studies, identified hyperactivity of the insula in patients with anxiety disorders compared to healthy controls, in response to stimulations of negative valence[9–12]. Preclinical studies have confirmed the implication of the insula in valence- and anxiety-related behaviors. Interestingly, according to the antero-posterior axis, the insula has been shown to have opposite control on positive and negative valence as well as anxiety-related behaviors, highlighting a functional dichotomy along its rostro-caudal axis. Specifically, optogenetic activation of excitatory neurons of the anterior or posterior insular cortices (aIC or pIC) respectively promotes approach and avoidance behaviors,

[1]Neurocentre Magendie, INSERM 1215, Université de Bordeaux, Bordeaux, France. [2]Research Group Synaptic Wiring and Information Processing, Center for Molecular Neurobiology Hamburg, University Medical Center Hamburg-Eppendorf, Hamburg, Germany. ✉e-mail: anna.beyeler@inserm.fr

suggesting the contribution of aIC in positive valence and pIC in negative valence[13,14]. However, a pharmacological study including both activation and inhibition of these insular regions during the elevated plus maze test (EPM) in mice, demonstrated that the aIC has anxiogenic properties, whereas the pIC has anxiolytic properties[15].

Multiple imaging studies, including a meta-analysis, also identified the amygdala as a key structure in patients with anxiety disorders[11]. In addition, the amygdala was shown to respond to stimuli of both positive and negative valence in healthy individuals[16], and the response to images of negative valence is heightened in patients with anxiety disorders[12]. Consistently, preclinical studies have reported that projection neurons of the basolateral amygdala (BLA) control anxiety-related behaviors[17,18], and that distinct neuronal populations of the BLA mediate positive and negative valence[19–21].

Alteration of the functional connectivity between the insula and the amygdala has been reported in patients with anxiety disorders, highlighting the potential contribution of this connection in the development of pathological anxiety[11,22,23]. Anatomical connections between these two regions have been described in mice[24–27], and optogenetic activation of aIC axonal fibers in the BLA imposes positive valence to a neutral stimulus, while reversing the aversive value of a bitter tastant in head-fixed mice[27] suggesting a role in positive valence coding. However, chemogenetic inhibition of aIC-BLA neurons prevents conditioned taste aversion, and chemogenetic activation of this population is sufficient to induce conditioned taste aversion[28], suggesting a role in negative valence coding. Moreover, most aIC-BLA neurons (80%) express serotonin receptors[29], which combined with the fact that the serotonergic system is a therapeutic target for anxiety disorders[30], suggests a contribution of aIC-BLA projection neurons in anxiety. However, the functional role of aIC-BLA neurons in anxiety- and valence- related behaviors remained unexplored. Using multifaceted circuit dissection in mice, including fiber photometry recording, anterograde tracing, electrophysiological mapping ex vivo, as well as pharmacological and optogenetic manipulations, we identified that aIC glutamatergic neurons are more active in anxiogenic environments and that this activity is positively correlated with the level of anxiety of animals. Interestingly, aIC-BLA neurons are partially responsible for this increased activity in anxiogenic spaces correlated with mice trait anxiety, and have anxiogenic properties. Additionally, aIC-BLA neurons bidirectionally respond to positive and negative stimuli, respectively with an inhibition and activation of their activity. Finally, we demonstrated that aIC-BLA level of activity while mice are in an anxiogenic space is positively correlated with their response immediately after an aversive stimulus, linking anxiety and negative valence within a single neural population of the insular cortex.

## Results

### Anterior but not posterior glutamatergic insula neurons increase their activity in anxiogenic spaces

To investigate the role of aIC and pIC in anxiety, we recorded the activity of projection neurons in these cortical regions during anxiety-related behaviors with fiber photometry, expressing the calcium sensor GCaMP6f under the CaMKIIα promoter (Fig. 1a, b and Supplementary Fig. 1a–d). In the EPM, we detected an increase of the global calcium signal in aIC, but not pIC excitatory neurons, while mice are exploring the open arms, in comparison to the closed arms (Fig. 1c, d, Supplementary Movie 1). Additionally, we observed that global calcium signal measured in the open arms is higher in aIC than pIC excitatory neurons (Fig. 1d). Similarly, during the open field test (OFT), aIC glutamatergic neurons, but not pIC neurons, exhibit an increase of calcium signal in the center compared to the borders of the arena (Fig. 1e, f). Moreover, the calcium signal in the center of the OFT is higher in aIC than pIC glutamatergic neurons (Fig. 1f). Thus, excitatory neurons in the aIC, but not the pIC, are activated during the exploration of anxiogenic spaces (open arms of the EPM and center of the OFT), although the overall

anxiety level of these mice is similar between the two groups (same time spent in anxiogenic spaces, Supplementary Fig. 1e, f). The increased activity in anxiogenic spaces is independent of locomotion, as the instantaneous velocity is higher in the anxiogenic space of the EPM (open arms), but higher in the safe space of the OFT (borders, Supplementary Fig. 1g, h). In addition, Pearson's correlation between the instantaneous velocity in the EPM and the OFT and the calcium signal recorded during the respective tests, shows $R^2$ inferior to 0.1 (Supplementary Fig. 1i) demonstrating the independence of locomotor velocity and calcium signal. Despite the lack of variation of the global calcium signal observed between the anxiogenic and the safe spaces for pIC glutamatergic neurons (Fig. 1d, f), the transient amplitude was lower in the open arms and the center of the EPM and OFT (Supplementary Fig. 1l, n) supporting previous findings[13].

As we found an increase in aIC excitatory neurons activity when mice are in anxiogenic spaces, we hypothesized these neurons encode anxiety-related information. In that case the activity of those neurons should increase as the animals reach the end of the open arms (Fig. 1g). We found that in both aIC and pIC, the global calcium signal is higher at the extremity compared to the beginning of the open arms, with a higher activity in the aIC compared to the pIC at the extremity of the open arms (Fig. 1h). In addition, we analyzed the calcium signal depending on both the position of the mouse in the open arms, and the movement direction within the open arms, averaging the signal when mice go out to explore the open arm (OUT), or back towards the center (BACK, Fig. 1i). Interestingly, the global calcium signal of aIC and pIC glutamatergic neurons is higher when mice are in the extremity compared to the beginning of the open arms, specifically while mice are going out in the open arms (Fig. 1j, k).

To evaluate the link between the activity of aIC projection neurons and trait anxiety, we correlated calcium signal in anxiogenic and safe spaces, with the overall anxiety level of individual animals, estimated by the percentage of time mice spent in anxiogenic spaces. Based on established predictive validity experiments, we know that the more anxious mice spend the least time in the EPM open arms and OFT center. Although, the global calcium signal recorded in the aIC while mice were located in anxiogenic spaces (EPM open arms and OFT center) did not correlate with the amount of time mice spent in the EPM open arms (Supplementary Fig. 1p, q), aIC neurons activity while mice were located in those anxiogenic spaces was strongly and negatively correlated with the time animals spent in the center of the OFT (Fig. 1l, n). Importantly, this correlation was absent for pIC excitatory neurons (Fig. 1m, o).

These results suggest that glutamatergic neurons in the aIC encode anxiogenic locations and trait anxiety. Thus, we hypothesized that inhibition of aIC glutamatergic neurons would modulate anxiety-related behaviors. Based on a previous anatomical study showing that the inhibitory 5-HT1A G-protein coupled receptors are expressed in 80% of aIC glutamatergic neurons[29], we used a pharmacological approach to inhibit aIC glutamatergic neurons by injecting the 5-HT1A agonist 8-OH-DPAT or the vehicle in the aIC before testing mice in the EPM (Fig. 1p–r, Supplementary Fig. 1r). Strikingly, inhibition of aIC glutamatergic neurons increases the time spent in the open arms compared to the vehicle control without influencing the locomotion (Fig. 1q, r).

Taken together, these results show that activity of glutamatergic neurons of the aIC encode anxiogenic spaces, are correlated with the level of trait anxiety, and control the level of anxiety-related behaviors.

### Anterior and posterior insula glutamatergic neurons are differentially active in response to negative and positive valence stimuli

Previous studies suggested that neurons in the aIC and pIC are involved in emotional valence processing[14], especially for the positive and negative valence of gustatory information, respectively. Thus,

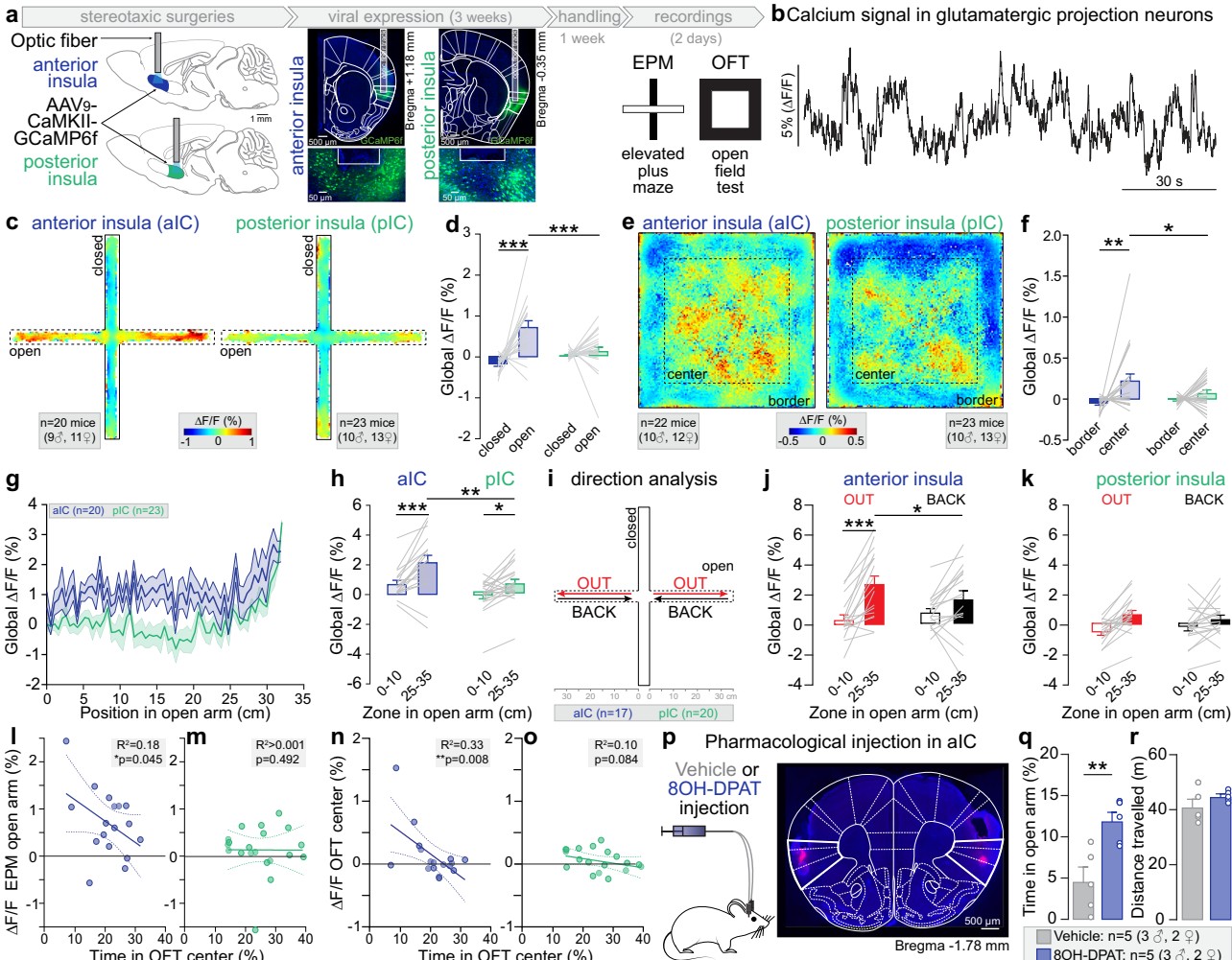

**Fig. 1 | Anxiety-related activity in aIC and pIC glutamatergic neurons. a** Strategy to record aIC or pIC glutamatergic neurons expressing GCaMP6f through a fiber implant. Representative confocal images of aIC and pIC neurons expressing GCaMP6f (in green). **b** Bulk GCaMP6f signal recorded from aIC glutamatergic neurons. ΔF/F represents the fluorescence changes from the mean level of the entire recording time series. **c** Averaged heat map of the calcium signal recorded from aIC and pIC neurons during 15 min of EPM test. **d** Average calcium signal recorded in aIC ($n = 20$ mice) and pIC ($n = 22$ mice) glutamatergic neurons in open and closed arms of the EPM (Two-way ANOVA, $F_{(1,41)} = 9.59$, **$p = 0.004$ for the brain region (aIC, pIC), $F_{(1,41)} = 22.07$, ***$p < 0.001$ for the zone (arms), $F_{(1,41)} = 13.19$, ***$p = 0.0008$ for region×zone interaction). The calcium signal is higher when mice are in the open compared to closed arms for aIC neurons (Bonferroni test ***$p < 0.0001$) and is higher for aIC than pIC neurons when mice are in the open arms (Bonferroni test ***$p < 0.001$). **e** Averaged heat map of the calcium signal recorded from aIC or pIC neurons during the OFT test. **f** Average calcium signal recorded in aIC ($n = 22$ mice) and pIC ($n = 23$ mice) glutamatergic neurons in the OFT border and center (Two-way ANOVA, $F_{(1,43)} = 11.36$, **$p = 0.002$ for the zone (arms), without effect of the brain region (aIC, pIC), and no interaction). The calcium signal in aIC neurons is higher when mice are in the center compared to the border (Bonferroni test ***$p = 0.002$), and is higher in aIC than pIC when mice are in the center (Bonferroni test *$p = 0.043$). **g** Average calcium signal recorded from aIC ($n = 20$ mice) and pIC ($n = 23$ mice) depending on the mice position in open arms of the EPM. **h** For both aIC and pIC neurons, the average calcium signal is higher when mice are at the end (25–35 cm) compared to the beginning (0–10 cm) of the open arms (Two-way ANOVA, $F_{(1,37)} = 29.60$, ***$p < 0.0001$ for the position, $F_{(1,37)} = 9.11$, **$p = 0.005$ for the brain region, without interaction; Bonferroni test for aIC ***$p < 0.0001$ and pIC *$p = 0.015$). The average calcium signal is higher in aIC than

pIC neurons when mice are at the end of the open arms (Bonferroni test **$p = 0.002$, $n = 18$ mice for aIC, $n = 21$ mice for pIC). **i** Calcium signal analysis during open arm exploration (OUT) and retreat (BACK) in the EPM. **j** In aIC neurons, the calcium signal is higher when mice are at the end compared to the beginning of the open arms, only for the OUT direction ($n = 17$ mice, Two-way ANOVA, $F_{(1,16)} = 19.69$, ***$p = 0.004$ for the position in the open arms, without effect of the direction and $F_{(1,16)} = 8.94$, **$p = 0.009$ for position x direction interaction; Bonferroni test for OUT ***$p < 0.0001$) and the calcium signal at the end of the open arms is higher for the OUT than the BACK direction (Bonferroni test *$p = 0.015$). **k** In pIC neurons, the signal is not different depending on the mouse direction ($n = 20$ mice, Two-way ANOVA, $F_{(1,19)} = 21.72$, ***$p = 0.0002$ for the position in the open arms, without effect of direction (OUT, BACK) and without interaction). **l** Calcium signal of aIC glutamatergic neurons when mice are in the open arms negatively correlates with the time mice spent in the OFT center (One-tailed Pearson correlation: $R^2 = 0.18$, *$p = 0.045$, $n = 17$ mice). **m** Correlation of pIC glutamatergic neurons calcium signal when mice are in the open arms, with the time mice spent in the OFT center (One-tailed Pearson correlation: $R^2 < 0.0001$, $p = 0.49$, $n = 20$ mice). **n** Calcium signal of aIC glutamatergic neurons in the OFT center correlates negatively with the time mice spent in the OFT center (One-tailed Pearson correlation: $R^2 = 0.33$, **$p = 0.008$, $n = 17$ mice). **o** Correlation of the calcium signal of pIC glutamatergic neurons when mice are in open arms with the time spent in the OFT center (One-tailed Pearson correlation: $R^2 = 0.10$, $p = 0.08$, $n = 20$ mice). **p** Representative image of a pharmacological injection in the aIC. **q** The time spent in the open arms is higher in mice who received 8-OH-DPAT intra-insula injection compared to the vehicle group (Two-tailed unpaired $t$-test, **$p = 0.008$, $n = 5$ mice). **r** Total distance travelled in the EPM is similar between 8-OH-DPAT and vehicle groups. All results are represented as mean ± SEM.

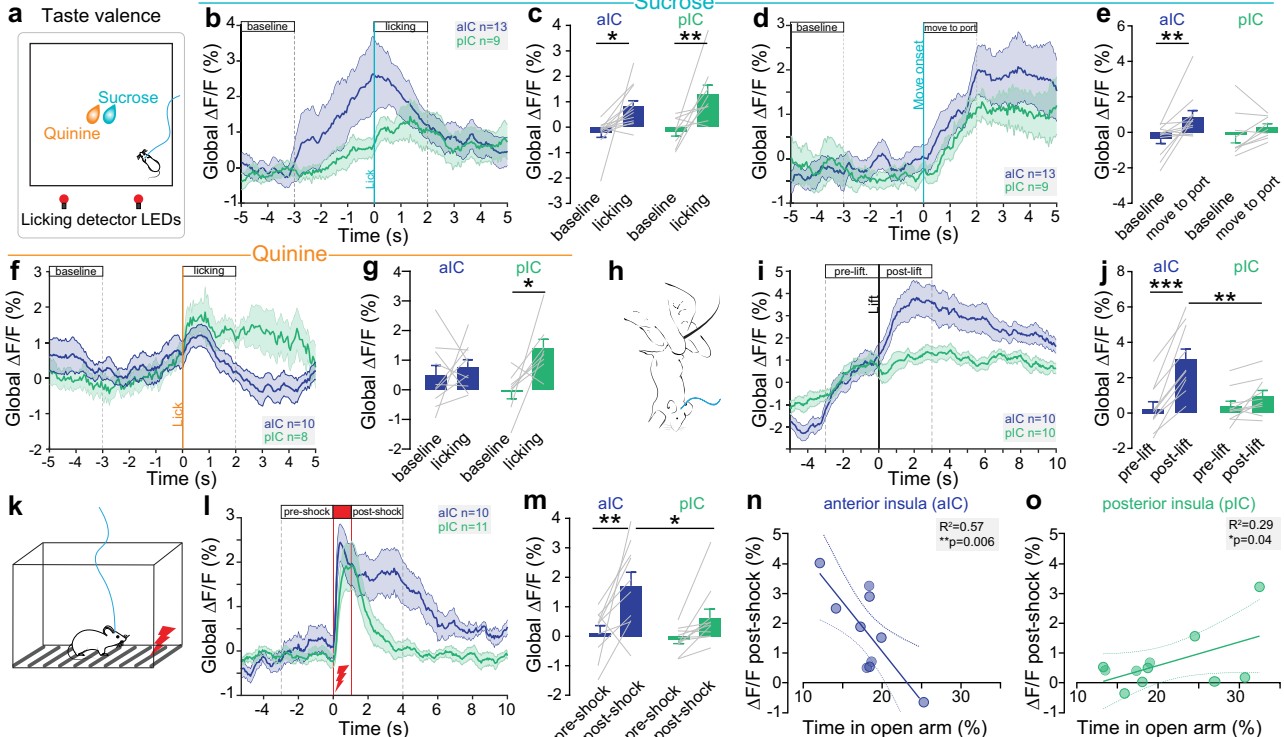

**Fig. 2 | Valence-related activity in aIC and pIC glutamatergic neurons.**
**a** Schematic of sucrose/quinine consumption test. **b** Peri-sucrose licking time course of calcium signal in the aIC ($n = 13$ mice) and pIC ($n = 9$ mice). **c** Calcium signal increases during sucrose licking in aIC ($n = 13$ mice) and pIC ($n = 9$ mice) glutamatergic neurons compared to baseline (Two-way ANOVA, $F_{(1,20)} = 19.57$, ***$p = 0.0003$ for the event (baseline, licking) with no effect of the region (aIC, pIC) and no interaction; Bonferroni test for aIC *$p = 0.017$, and for pIC **$p = 0.007$).
**d** Peri-event analysis of the calcium signal between baseline and movement onset toward the sucrose port in the aIC and pIC (aIC $n = 13$ mice, pIC $n = 9$ mice).
**e** Calcium signal increased after movement towards the sucrose port in aIC neurons compared to baseline without changes for pIC neurons ($n = 13$ and $n = 9$ mice, Two-way ANOVA, $F_{(1,20)} = 8.802$, **$p = 0.008$ for the event (baseline, move to port) with no effect of the region (aIC, pIC) and no interaction; Bonferroni test for aIC **$p = 0.007$). **f** Peri-quinine licking time course of calcium signal in the aIC ($n = 10$ mice) and pIC ($n = 8$ mice). **g** Calcium signal increases during quinine licking in pIC neurons ($n = 8$ mice), but not aIC neurons ($n = 10$ mice) compared to baseline (Two-way ANOVA, $F_{(1,16)} = 6.957$, *$p = 0.018$ for the event (baseline, licking), with no effect of region and no interaction; Bonferroni test for pIC *$p = 0.019$). **h** Schematic of the

tail suspension test. **i** Peri-event calcium signal in the aIC ($n = 10$ mice) and pIC ($n = 10$ mice) before and during tail suspension. **j** Calcium signal increases post-lift in aIC ($n = 10$ mice), but not pIC ($n = 10$ mice) neurons, compared to pre-lift (Two-way ANOVA, $F_{(1,18)} = 49.54$, ***$p < 0.001$ for the event (pre- vs post-lift), no effect of the region (aIC, pIC), and $F_{(1,18)} = 21.04$, ***$p = 0.0002$ for event x region interaction; Bonferroni test for aIC, ***$p < 0.0001$) and is higher in aIC compared to pIC post-lift (Bonferroni test **$p = 0.003$). **k** Schematic of the footshocks test. **l** Peri-shock time course of calcium signal in the aIC ($n = 10$ mice) and pIC ($n = 11$ mice). **m** Calcium signal increases post-shock in aIC ($n = 10$ mice), but not pIC ($n = 11$ mice) neurons compared to pre-shock (Two-way ANOVA, $F_{(1,19)} = 17.84$, ***$p = 0.005$ for the event (pre- vs post-shock) with no effect of the region (aIC, pIC) and no interaction; Bonferroni test for aIC **$p = 0.002$), and is higher in aIC compared to pIC post-shock (Bonferroni test *$p = 0.032$). **n** Post-shock calcium signal in aIC neurons negatively correlates with the time mice spent in open arms of the EPM (One-tailed Pearson correlation: $R^2 = 0.57$, **$p = 0.006$, $n = 10$ mice). **o** Post-shock calcium signal in pIC neurons positively correlates with the time mice spent in open arms of the EPM (One-tailed Pearson correlation: $R^2 = 0.29$, *$p = 0.043$, $n = 11$ mice). All the results are represented as mean ± SEM.

we performed fiber photometry recordings of aIC and pIC glutamatergic neurons during sucrose and quinine consumption (Fig. 2a, b) to evaluate their dynamics in response to gustatory positive and negative valence. As sucrose is appetitive and quinine aversive to mice, the number of licks and number of licking bouts are higher for sucrose than for quinine in both aIC and pIC groups (Supplementary Fig. 1s, t). Peri-licking analysis of the calcium signal shows that aIC and pIC excitatory neurons activity equally increases during sucrose licking compare to baseline (Fig. 2c). As the increase of signal appears to start before the onset of licking for both aIC and pIC (Fig. 2b), we analyzed the neural activity starting from the mouse movement initiation towards the sucrose port, and observed an increase activity for aIC, but not for pIC glutamatergic neurons, compared to baseline (Fig. 2d, e), suggesting an anticipatory activity of aIC glutamatergic neurons for positive valence. During quinine licking, increase of calcium global signal is detected in pIC, whereas no significant change is observed in aIC excitatory neurons compared to baseline (Fig. 2f, g). Altogether, these recordings suggest that aIC preferentially encodes gustatory positive stimuli whereas pIC glutamatergic neurons respond to both

positive and negative stimuli which is consistent with the literature[31]. To extend these results to non-gustatory stimuli, we recorded aIC and pIC glutamatergic neurons in response to two other negatively-valenced stimuli, namely tail suspensions and mild footshocks (Fig. 2h–m). During tail suspension the calcium signal strongly increases compared to the pre-lift period, specifically in aIC glutamatergic neurons (Fig. 2i, j). Similarly, after footshock, the calcium signal increases compared to the pre-shock time window selectively in aIC glutamatergic neurons (Fig. 2k–m). Importantly, aIC glutamatergic neurons activity post-shock is strongly and negatively correlated with the time animal spent in the open arms of the EPM (Fig. 2n). On the contrary, pIC glutamatergic neurons activity post-shock is positively correlated with the time mice spent in the open arms of the EPM (Fig. 2o). Consistently, during tail suspension, aIC neural activity tends to be negatively correlated with the time mice spent in the open arms, while pIC neural response is positively correlated with this estimate of trait anxiety (Supplementary Fig. 2e, f).

Altogether, the activity of both aIC and pIC excitatory neurons increase after a positive gustatory stimulus, but only pIC neurons are

activated in response to a negative gustatory stimulus. Conversely, only aIC glutamatergic neurons are activated by non-gustatory negative stimuli, and this response intensity is linked to the level of anxiety of the animals. Finally, the opposite correlations found in aIC and pIC between neural activity in response to an aversive stimulus and anxiety-related behaviors, pinpoint the existence of a functional antero-posterior dichotomy of the neural link between anxiety and valence in the insula.

As the insular cortex is known to be composed of functionally divergent neural populations[24,29], we undertook to anatomically characterize projections patterns of aIC glutamatergic neurons to other key brain regions involved in anxiety or valence.

## Mapping of anterior insula projection neurons and aIC-BLA collateralization

To map the density of long-range projections of glutamatergic insula neurons, we virally expressed eYFP in glutamatergic neurons of the aIC or pIC to label their cell bodies, dendrites, and axonal projections (Fig. 3a, Supplementary Fig. 3a–d). After 4 weeks of expression, eYFP fluorescence was imaged in twelve downstream regions (Fig. 3b, c, Supplementary Fig. 3e, f), quantified and normalized to the BLA as the densest axonal density arising from the aIC is detected in the BLA (Fig. 3d, e). Importantly, a substantial amount of axonal fibers from the aIC is also detected in the lateral and in the medial section of the central amygdala (CeL and CeM), and axonal fibers from the pIC are also detected in the BLA, which challenges the notion of two segregated insula-amygdala pathways (aIC-BLA and pIC-CeA)[27]. In addition, we identified strong contralateral projections from the right-to-left aIC, and right-to-left pIC, as well as dense axonal fibers from both aIC and pIC in the nucleus accumbens core (NAc, Fig. 3d, e, Supplementary Fig. 3f, g). The dual projection of aIC and pIC to the basolateral and central amygdala nuclei, as well as the selectivity of projection to the controlateral insula are represented in a summary diagram (Fig. 3e).

As the BLA is the main output of the aIC, we further dissected the distribution of collaterals arising from the aIC-BLA pathway (Fig. 3f–k, Supplementary Fig. 3h–j). We used a triple viral strategy to quantify the axonal projections as well as the synaptic contacts of aIC-BLA neurons across eleven brain regions (Fig. 3f). We injected a retrograde viral vector expressing the cre-recombinase in the BLA and a mix of two cre-dependent adeno-associated viral vectors in the aIC to express the yellow fluorescent protein eYFP in all neural processes, and to express the synaptic protein Synaptophysin, fused to the red fluorescent protein mCherry (SynP-mCh), in the synaptic boutons (Fig. 3f). We found that on average, the collateral density of aIC-BLA neurons is similar in the ipsilateral CeL and NAc, relative to the ipsilateral BLA (Fig. 3i). Interestingly, in the contralateral side, the axonal density is the highest in the BLA and aIC, suggesting that bilaterally, the main output of aIC-BLA neurons are the BLA and aIC, while ipsilaterally, other nuclei are also strongly innervated (CeL and NAc). Consistently, SynP-mCh labeling of synapses originating from aIC-BLA neurons confirmed that the densest contacts are in the ipsilateral BLA, CeL, and NAc as well as the contralateral BLA and aIC (Fig. 3j).

Taken together, this anatomical study highlights that aIC glutamatergic neurons mainly project to the BLA, and that aIC-BLA neural population is more complex than a direct pathway and recruits a whole circuit including important bilateral collateralization (Fig. 3k).

## Synaptic and cellular properties of insula to BLA and CeM pathways

The insula innervates both the BLA, main input nucleus of the amygdala, and the CeM, main output structure of the amygdala. While the BLA is the nucleus receiving the densest insular projection, the CeM is the amygdala nucleus receiving the lowest density of aIC axons (Fig. 3d) and of aIC-BLA collaterals (Fig. 3i, j). Considering those anatomical divergences, we tested whether these two parallel insula-

amygdala pathways have different features by recording electro-physiological properties of IC-BLA and IC-CeM synapses and cell bodies.

First, to validate the existence of a direct monosynaptic connection from insula neurons onto amygdala neurons, we performed optogenetic circuit mapping of insula synaptic inputs onto BLA and CeM neurons using whole-cell patch-clamp recordings[17,32]. We injected AAV$_9$-CaMKII$\alpha$-ChR2-eYFP (ChETA 2.0 variant for ultrafast optogenetic control) in the IC to record optically evoked excitatory and inhibitory postsynaptic currents (oEPSCs and oIPSCs) in neurons of amygdala nuclei clamped at −70 mV and 0 mV, respectively (2 ms light pulse, Fig. 4a and Supplementary Fig. 4a–c). As expected, BLA neurons have a higher membrane capacitance and a lower membrane resistance compared to CeM neurons, consistent with larger BLA excitatory neurons and smaller CeM inhibitory neurons (Supplementary Fig. 4a). In both BLA and CeM neurons, a fraction of oEPSCs remained after blockade of network activity (TTX + 4AP, Fig. 4b, c), indicating monosynaptic excitatory inputs from IC on both BLA and CeM neurons. Elimination of monosynaptic excitatory responses by glutamatergic antagonists (AP5 + NBQX) confirmed their glutamatergic nature (Fig. 4b, c). Interestingly, oIPSCs are systematically present, and abolished after network activity blockade, meaning they are polysynaptic. Optogenetic currents latency are twice longer for oIPSCs than for oEPSCs in both IC-BLA and IC-CeM synapses (Fig. 4d). Altogether, these circuit mapping experiments demonstrate that IC glutamatergic neurons mono- and poly-synaptically excite BLA and CeM neurons, and polysynaptically recruit local inhibition.

Second, to examine the short term dynamics of insula synapses in the BLA and CeM, we used paired- or train-pulse opto-stimulation protocols[33]. IC-BLA and IC-CeM excitatory and inhibitory paired-pulse ratios (PPRs) are <1, indicating they both have depressing properties (Fig. 4e). Conversely, train stimulations revealed that starting from the third photostimulation, IC-BLA synapses are more depressed than IC-CeM ones (Fig. 4f, g).

Finally, as IC-BLA and IC-CeM synapses exhibit different properties, we hypothesize that the IC-BLA and IC-CeM cell bodies in the insula also have different intrinsic properties, and recorded them thanks to retrograde tracer injections in the BLA and CeM (Fig. 4h-n and Supplementary Fig. 4e-g). The analysis of intrinsic properties of IC-BLA and IC-CeM neurons depending on their antero-posterior location, revealed that aIC-BLA neurons have a lower membrane and input resistance than pIC-BLA neurons, as well as a higher rheobase, suggesting pIC-BLA are more excitable (Fig. 4k). Given that intrinsic properties of pyramidal neurons in other cortices show interlaminar differences[34], we also analyzed IC-BLA and IC-CeM neurons depending on their laminar location, and found that membrane capacitance of IC-BLA neurons is higher in layer 5, compared to layer 2/3 (Fig. 4l). Moreover, aIC-CeM neurons fire at a higher frequency than pIC-CeM neurons with the same amount of current injected (Fig. 4m, n). Finally, when pooling data across the antero-posterior and mediolateral axes, we found that IC-BLA neurons have a higher membrane capacitance than IC-CeM neurons, suggesting IC-BLA neurons are larger (Supplementary Fig. 4e). Other passive properties, such as membrane and input resistance, are comparable between these two populations of projection neurons (Supplementary Fig. 4e). However, the active response to current injection is higher in IC-CeM neurons for low injected currents (40 to 80 pA, Supplementary Fig. 4f).

Overall, this electrophysiological dissection reveals that insula neurons have different electrical and synaptic properties depending on their projection target within the amygdala, suggesting they might support different functions.

## aIC-BLA projection neurons control anxiety and are more active in anxiogenic spaces

As showed that glutamatergic neurons of the aIC are more active in anxiogenic spaces (Fig. 1), and their main downstream target is the BLA

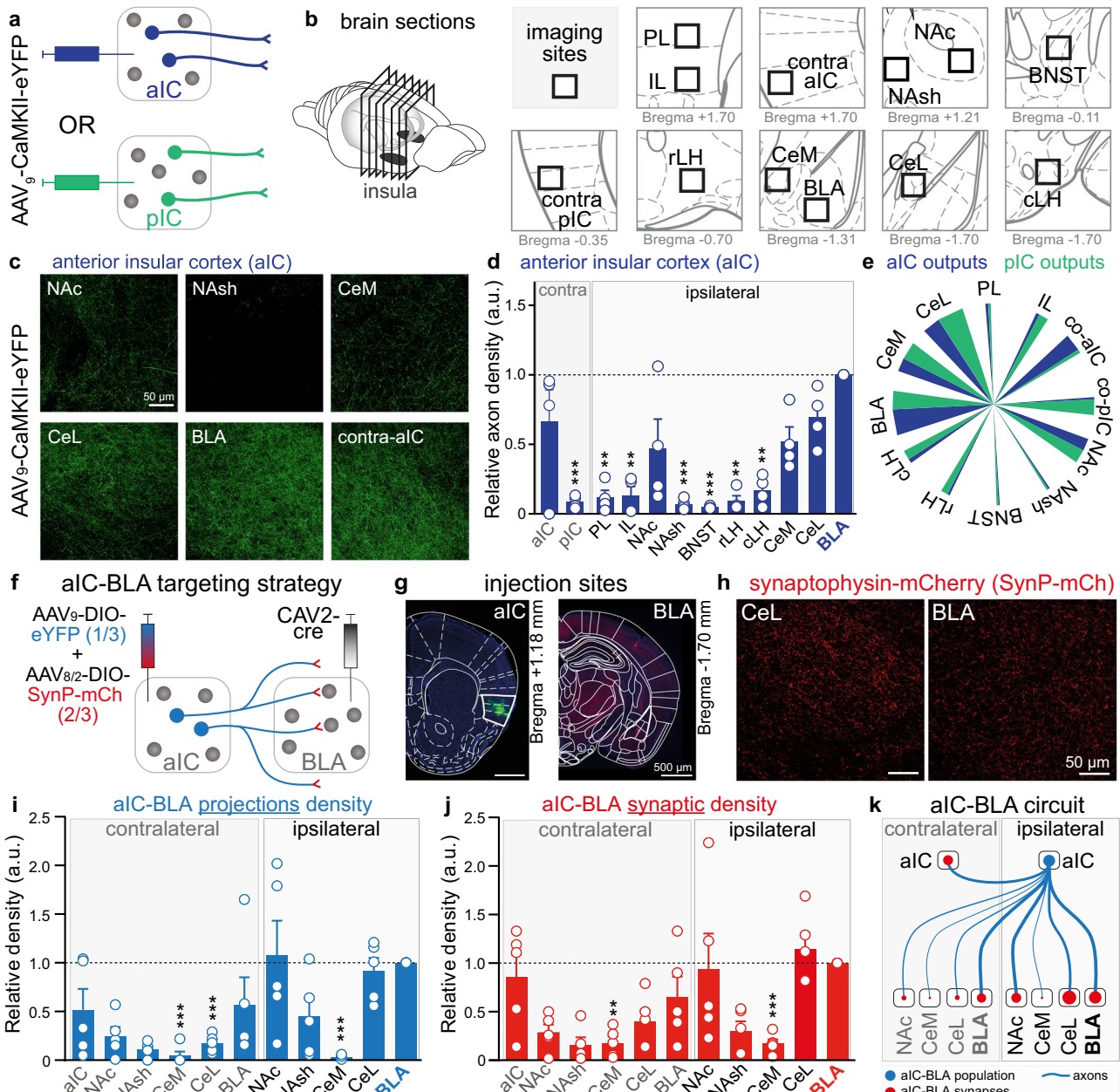

**Fig. 3 | Downstream projections of anterior insula neurons and aIC collaterals. a, b** Experimental scheme of viral expression **a** and imaging downstream regions **b**, including prelimbic (PL), infralimbic (IL) cortices of the medial prefrontal cortex, contralateral aIC (contra-aIC), contralateral pIC (contra-pIC), nucleus accumbens core (NAc), nucleus accumbens shell (NAsh), bed nucleus of the stria terminalis (BNST), rostral lateral hypothalamus (rLH), caudal lateral hypothalamus (cLH), basolateral amygdala (BLA), medial (CeM) and lateral (CeL) subdivisions of central amygdala. **c** Representative images of axonal projections from one mouse expressing eYFP in glutamatergic aIC neurons. **d** Number of fluorescent pixels normalized to the average value of the maximal projection region (BLA) per image from aIC neurons (*n* = 4 mice). The two bar graphs on the left represent the contralateral aIC and pIC relative axonal density (One-way ANOVA, *\*p* = 0.0198, Bonferroni test compared to BLA image \*\**p* < 0.01, \*\*\**p* < 0.001). **e** Summary pie chart of relative fluorescent intensity in projecting regions from aIC (blue) and pIC (green).

**f** Viral vector strategy to target aIC to BLA collaterals with eYFP labelling the axonal density and synaptophysin-mCherry (SynP-mCh) targeting synaptic inputs. **g** Representative images of aIC (left) and BLA (right) transfected with a cre-dependent dual-viral vector and CAV2-cre respectively. **h** Representative confocal images of SynP-mCherry in the CeL and the BLA. **i, j.** Number of fluorescent pixels representing aIC to BLA axonal **i** and synaptic **j** density normalized to the average value of the ipsilateral BLA image (*n* = 5 mice). (One-way ANOVA, **i** *\*p* = 0.0193, Bonferroni test compared to ipsi-BLA image \*\**p* < 0.01, \*\*\**p* < 0.001; **j** \*\**p* = 0.0095, Bonferroni test compared to ipsi-BLA image \*\**p* < 0.01, \*\*\**p* < 0.001). **k** Summary of the aIC-BLA circuit including ipsilateral and contralateral projections. Blue represents the aIC-BLA population and the red dots represent aIC-BLA synapses. The width of the lines is proportional to the axonal density and the size of the red dots is proportional to the number of synaptic inputs. All the results are represented as mean ± SEM.

(Fig. 3), we hypothesized that aIC-BLA neurons are a major contributor to this anxiety-linked activity in the insula. To test the causal role of aIC-BLA projection neurons in anxiety-related behaviors, we used an optogenetic approach during anxiety assays, using the novel opsin somBiPOLES[35]. This soma-targeted opsin is a fusion protein of the inhibitory opsin *Gt*ACR2[36] and the excitatory opsin Chrimson[37], enabling activation and inhibition of the same neuronal populations through illumination at different wavelengths (Fig. 5a–c). After ex vivo patch-clamp validation (Supplementary Fig. 5e–i), we expressed som-BiPOLES bilaterally, in aIC-BLA neurons through a dual viral vector

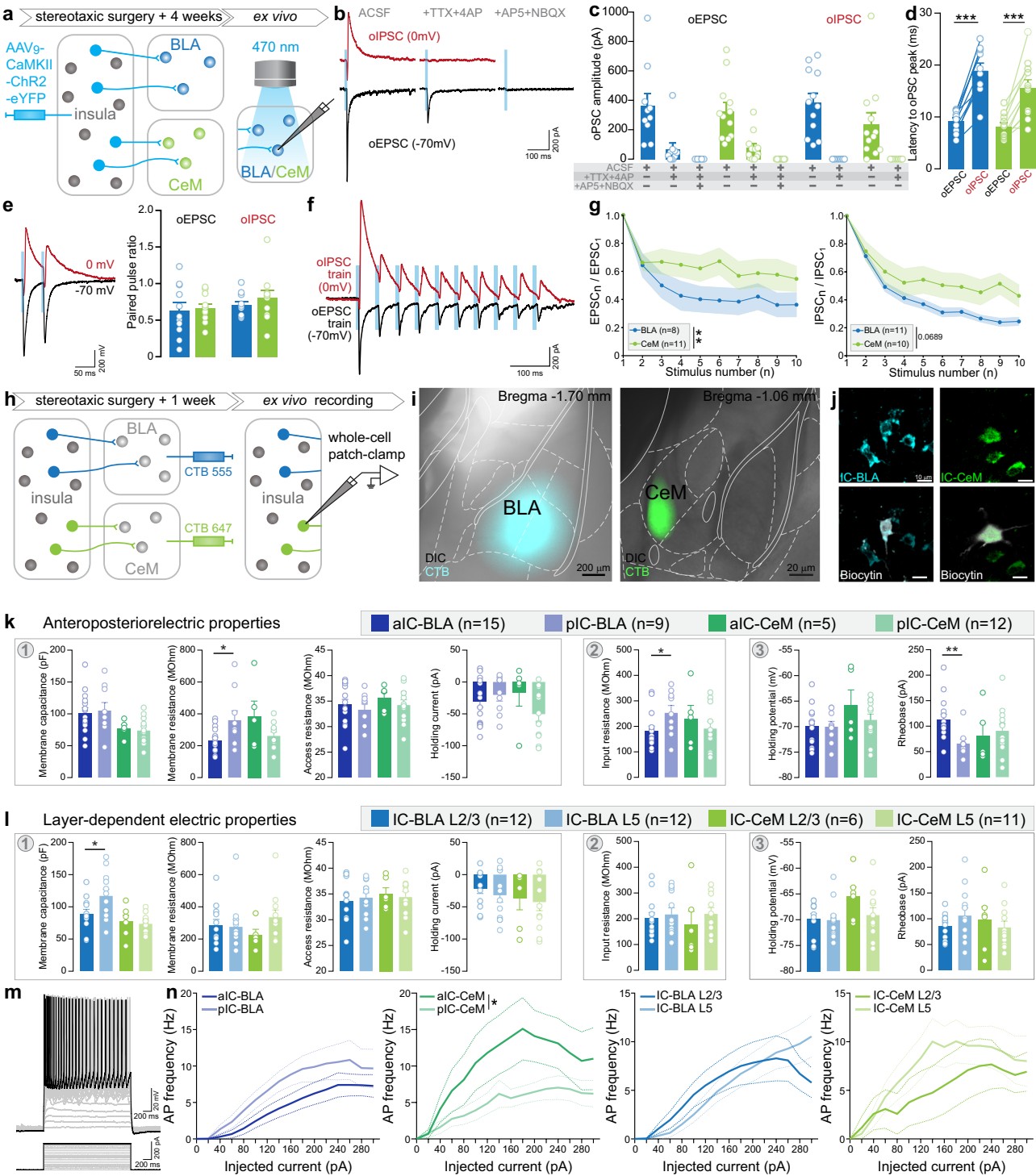

approach and manipulated their activity through optic fibers implanted above the aIC (Supplementary Fig. 5a, b). To evaluate the instantaneous effect of activation or inhibition of aIC-BLA neurons on anxiety-related behaviors, mice were tested in sessions composed of 6 epochs, beginning with neural activation (orange light), followed by inhibition (blue light) and a resting epoch (OFF, Fig. 5d, e). Averaged over all epochs (activation/inhibition/OFF), the time spent in the anxiogenic zone, is lower for the somBiPOLES group, which spent less time in the center of the OFT and tend to spend less time in the open arms of the EPM (*p* = 0.23), in comparison to the control group (mCerulean, Fig. 5d, e). Importantly, no effect of light is detected on

locomotion, as measured by distance travelled in the OFT (Fig. 5f). After behavioral tests, aIC-BLA neurons were illuminated with orange light (activation of Chrimson), and immunofluorescent staining of cFos in fixed brain slices revealed a significant increase of cFos expressing cells in somBiPOLES expressing neurons compared to control neurons expressing mCerulean (Supplementary Fig. 5c, d). Taken together, our results suggest that aIC-BLA have a functional role in anxiety-related behaviors. Nevertheless, as both activation and inhibition of aIC-BLA projections neurons decrease anxiety-related behavior, these causal experiments do not provide information on how aIC-BLA neurons encode anxiety.

**Fig. 4 | Properties of IC-BLA and IC-CeM synapses, and of IC-BLA and IC-CeM neurons. a** Experimental strategy of expressing ChR2-eYFP in the insular neurons to confirm the monosynaptic connection of the insular neurons to basolateral amygdala (BLA, IC-BLA) and medial subdivision of central amygdala (CeM, IC-CeM) and record the dynamic properties of insular synapses on BLA and CeM neurons upon the optogenetic activation of insular terminals. **b** Representative traces of synaptic responses upon the application of ACSF, +TTX + 4AP and +AP5 + NBQX in order to prove the monosynaptic connection of the insular neurons to the BLA and CeM neurons. **c** Quantification of optogenetic-induced excitatory/inhibitory post-synaptic currents (oEPSC/oIPSC) of BLA and CeM neurons upon optogenetic activations of insular axonal terminals during the application of ACSF, +TTX + 4AP and +AP5 + NBQX (Two-way ANOVA: drug effect for oEPSC: $F_{(2,40)} = 44.08$, $p < 0.0001$ from BLA $n = 10$ cells, CeM $n = 12$ cells; drug effect for oIPSC: $F_{(1,22)} = 39.34$, $p < 0.0001$ for BLA $n = 12$ cells, CeM $n = 12$ cells. **d** Latency to the peak of oEPSC or oIPSC (Two-tailed paired $t$-test, t = 9.111 ***$p < 0.0001$ for BLA $n = 10$ cells, t = 5.467, ***$p = 0.0003$ for CeM $n = 11$ cells). **e** Representative traces and summary data of paired pulse ratio at -70 mV (excitatory PPR) and 0 mV (inhibitory PPR) at the IC-BLA and IC-CeM synapses. (oEPSC: BLA $n = 11$ cells, CeM $n = 12$ cells, oIPSC: BLA $n = 11$ cells, CeM $n = 11$ cells). **f, g** Representative traces **f** and summary data **g** of excitatory or inhibitory responses of BLA and CeM neurons upon the 10 train stimulations of ChR2 in insular terminals (Repeated measures of ANOVA: Interaction effect for EPSC: $F_{(9,153)} = 2.563$, **$p < 0.01$, Interaction effect for IPSC: $F_{(9,180)} = 1.843$, $p = 0.0634$). **h.** Experimental plan for CTB labeling and whole-cell patch recording of IC-BLA and IC-CeM. **i.** Representative images of CTB injection sites in BLA and CeM. **j.** Representative images of biocytin-filled neurons labelled by CTB. **k, l** Analysis of intrinsic properties from IC-BLA and IC-CeM neurons depending on their anterior-posterior **k** and layer **l** position. Measures were obtained from the membrane seal test [1], the IV curve [2], and the Ramp test [3]. See Supplementary Fig. 4a for example traces. **k** Membrane and input resistances were higher for pIC-BLA compared to aIC-BLA neurons (Two-tailed $t$-test, t = 2.415, *$p = 0.025$, and t = 2.154, *$p = 0.042$ respectively). The rheobase was higher in aIC-BLA neurons (Two-tailed $t$-test, t = 3.164, **$p = 0.005$). For aIC-BLA $n = 15$ cells, pIC-BLA $n = 9$ cells, aIC-CeM $n = 5$ cells and pIC-CeM $n = 12$ cells. **l** Membrane capacitance is higher for L5 than L2/3 IC-BLA neurons (Two-tailed $t$-test, t = 2.219, *$p = 0.037$). For IC-BLA L2/3 and L5 $n = 12$ cells, IC-CeM L2/3 $n = 6$ cells and IC-CeM L5 $n = 11$ cells. **m** Representative trace of output firing in response to input current steps, analyzed in the following panel. **n** aIC-CeM neurons are more excitable than pIC-CeM neurons (Two-way ANOVA, $F_{(1,15)} = 5.958$, *$p = 0.023$). All results are represented as mean ± SEM.

Thus, we used fiber photometry during anxiety-related behaviors, by expressing GCaMP6m selectively in aIC-BLA neurons through a cre-dependent dual-viral strategy, and implantation of an optical fiber in the aIC (Fig. 5g, h and Supplementary Fig. 6a–c) to record calcium signals of aIC-BLA neurons (Fig. 5i, j). While mice explored the EPM (Fig. 5k), the global calcium signal and the frequency of calcium transients increased in the open arms compared to the closed arms (Fig. 5l, m and Supplementary Movie 2). Notably, velocity in the open and closed arms are similar (Supplementary Fig. 6d) and aIC-BLA activity do not correlate with instantaneous velocity during the EPM test (Supplementary Fig. 6e). In the OFT, we also observe an increase of the calcium global signal in the anxiogenic space (center, Fig. 5n, o), as well as a trend for an increased frequency of calcium transients ($p = 0.052$, Fig. 5p). While the velocity is higher in the center compared to the border of the OFT (Supplementary Fig. 6d), aIC-BLA activity does not correlate with movement velocity during the OFT test (Supplementary Fig. 6e).

Interestingly, the global signal of aIC-BLA neurons is increased at the end of the open and closed arms compared to the beginning but with a higher amplitude in the open arms (Fig. 5q, r). Moreover, the activity of aIC-BLA neurons increases at the extremity of the open arms when the mice travel OUT toward the end but not when going BACK to the center of the maze (Fig. 5s, t). These results suggest that aIC-BLA neurons encode information relative to anxiogenic spaces.

To evaluate the role of aIC-BLA projection neuron activity and trait anxiety, we correlated the difference between the calcium transient frequency in anxiogenic and safe spaces, with the overall anxiety level of individual animals, estimated by the percentage of time spent in the open arms of the EPM (most anxious mice spend the least time in the open arms). Interestingly, the differential transient frequency between the anxiogenic and safe spaces of the EPM (open-closed) is positively correlated with the anxiety level of the animals (Fig. 5u), linking the transient activity of aIC-BLA neurons in anxiogenic spaces to the animal level of trait anxiety. Altogether, our data show that the activity of aIC-BLA neurons is increased in anxiogenic spaces, encodes state anxiety, and is correlated with trait anxiety.

### Bidirectional representation of valence in aIC-BLA projection neurons

To test the causal role of aIC-BLA projection neurons in valence-related behaviors, we used a cre-dependent dual-viral strategy to express the inhibitory opsin *Gt*ACR2 in aIC-BLA neurons and implanted an optic fiber over the aIC (Supplementary Fig. 7a-f). After confirming that illumination of aIC-BLA neurons expressing *Gt*ACR2 induces an inhibition of action potential firing in these neurons using whole-cell patch-clamp recordings ex vivo (Supplementary Fig. 7a-d), we tested the impact of inhibition of aIC-BLA neurons in a closed-loop real-time place preference/avoidance assay (RTPP/A). In this test, mice freely explored two chambers, including one where aIC-BLA projection neurons were inhibited (Fig. 6a). Photo-inhibition of aIC-BLA glutamatergic neurons induces a preference for the light-paired side compared to control mice (Fig. 6b). In an independent group of mice, photo-activation of aIC-BLA neurons by expressing ChR2 (Supplementary Fig 7g, h) does not induce significant behavioral changes, although we observe a trend towards a preference for the light-paired side, compared to control mice (Supplementary Fig. 7i, j, $p = 0.07$). Together, these data show that inhibition of aIC-BLA projection neurons can induce place preference, suggesting that these neurons encode negative valence.

To identify how aIC-BLA neurons encode emotional valence, we performed fiber photometry recordings of this neuronal population during valence-related behaviors, including sucrose and quinine consumption. As expected, the number of licks and licking bouts for sucrose is higher than for quinine consumption (Supplementary Fig. 6f). During sucrose consumption (Fig. 6c), we observe a decrease in the calcium signal after the onset of sucrose licking (Fig. 6d, e). In contrast, quinine consumption does not alter calcium signal after the onset of licking (Fig. 6d,f). However, tail suspension and mild foot-shocks, two stimuli of negative valence, strongly increase aIC-BLA calcium signal (Fig. 6g, h, j, k). Importantly, aIC-BLA calcium signal during tail suspension is negatively correlated with the time mice spent in the open arms of the EPM (Fig. 6i), suggesting this pathway could be a neurobiological link between negative valence and trait anxiety. Interestingly, the responses of aIC-BLA neurons to sucrose and to tail suspension are negatively correlated (Supplementary Fig. 8h) supporting a valence selectivity. Finally, calcium signal of aIC-BLA in the open arms of the EPM correlates positively with the calcium global signal during the post-shock period (Fig. 6l), pinpointing aIC-BLA neurons as a biological link between negative valence and anxiety.

Together, these data suggest that aIC-BLA neurons bidirectionally encode valence through an inhibition to positive valence and activation to negative valence. Our results also support a model where the aIC-BLA pathway is, at least in part, a neurobiological link between anxiety- and valence-related behaviors.

## Discussion

Using fiber photometry recordings, pharmacological and optogenetic manipulations, as well as anatomical and electrophysiological

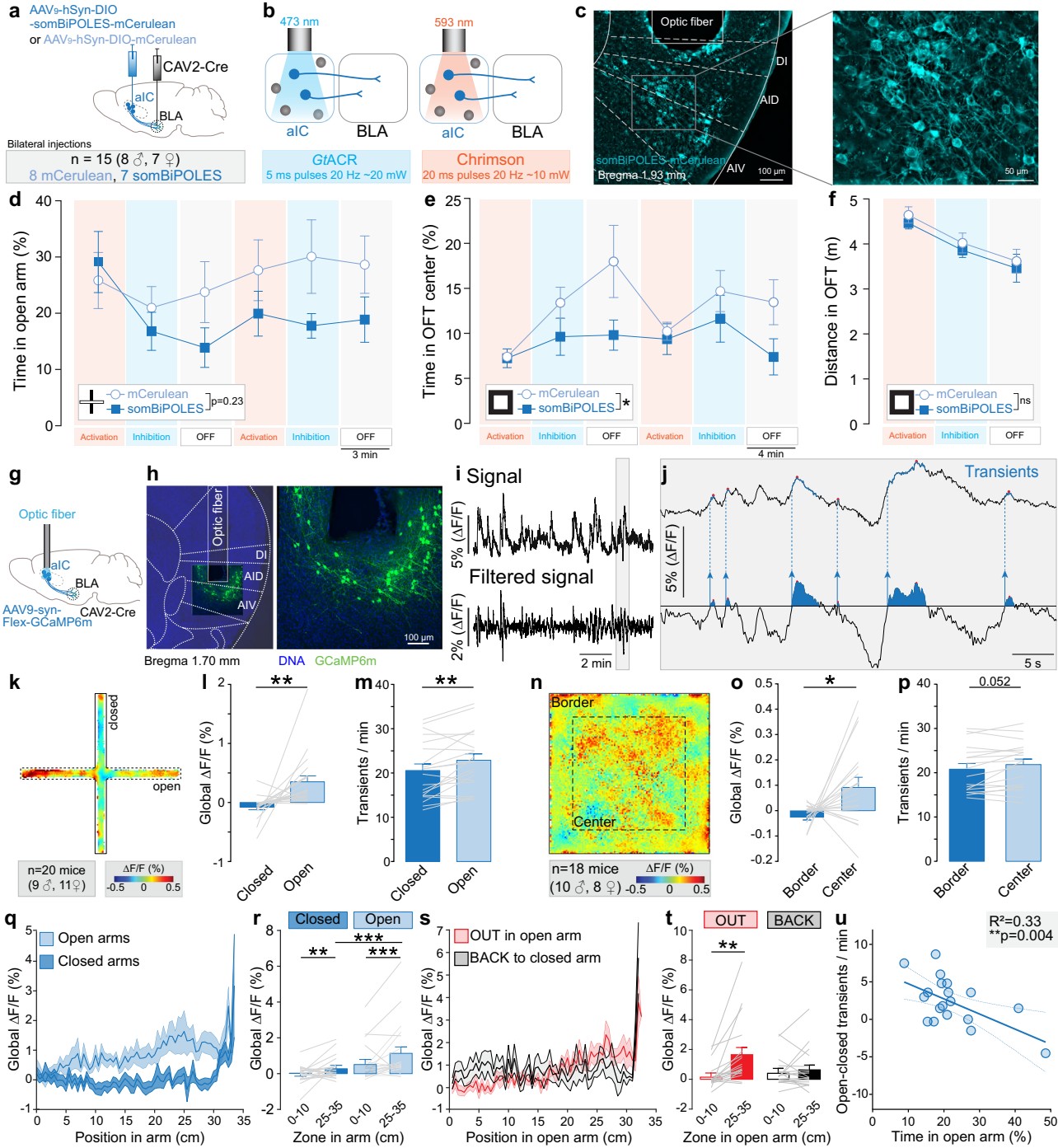

## Functional antero-posterior dichotomy of the insula in anxiety and emotional valence

mapping of insular cortex neurons our study uncovers aIC, and more specifically aIC-BLA neurons as a common neural substrate for emotional valence and anxiogenic spaces.

### Functional antero-posterior dichotomy of the insula in anxiety and emotional valence

In vivo calcium imaging using fiber photometry showed that glutamatergic neurons of the aIC are selectively more active in anxiogenic spaces, as we did not observe this increased activity in the pIC. Recent work from others, had identified a contribution of the pIC in processing aversive states and anxiety-related behaviors[13]. Interestingly, this study described a decrease in pIC glutamatergic neurons activity when mice were located in the open arms of elevated mazes. In our experimental conditions, we observed moderate and variable changes in pIC

neuronal activity in the EPM. This discrepancy might be explained by the fact that most of their recordings were performed in mice exploring an elevated zero maze (EZM, $n = 8$, EPM $n = 3$)[13]. Mice spend twice more time in the open arms of EZM than of the EPM, demonstrating that the EZM is less anxiogenic than the EPM[38,39]. Although our experimental design did not capture a decrease of pIC global calcium signal in the open arms of the EPM, we observed that calcium transients amplitude decreased in the open compared to the closed arms (Supplementary Fig. 1l, n) which is consistent with this previous finding[13].

Importantly, we also observed a negative correlation of aIC calcium signal in anxiogenic spaces of the EPM and OFT, with the time animals spent in the center of the OFT, which was selective of the anterior insula, as both correlations were absent in the posterior insula.

**Fig. 5 | Role of aIC-BLA neurons during anxiety-related behaviors.**
**a** Experimental design including an injection of a cre-dependent viral vector to express somBiPOLES-mCerrulean (or mCerulean) in cell bodies of the aIC, and the injection of the CAV2-Cre vector in the BLA. **b** Pattern of somBiPOLES experimental manipulation, in which blue light (473 nm) is used to inhibit neuronal population through the soma-targeted *Gt*ACR2 component of somBiPOLES, while orange (593 nm) light is used to stimulate neuronal population through the Chrimson component of somBiPOLES. **c** Representative images of a coronal brain section containing aIC neurons expressing somBiPOLES below the fiber implant. **d** Percentage of time spent in open arms of EPM, across the 6 epochs of 3 mins. Independently of the stimulation epoch procedure, somBiPOLES group tend to spend less time in open arms compared to control mCerulean (Two-way ANOVA, time: $F_{(3.061, 39.79)} = 2.28$, $p = 0.09$, opsin: $F_{(1,13)} = 1.596$, $p = 0.228$, time x opsin interaction: $F_{(5,65)} = 1.610$, $p = 0.170$, somBiPOLES $n = 7$ mice, mCerulean $n = 8$ mice). **e** Percentage of time spent in the center of OFT, across each epoch. Independently of the stimulation epoch procedure, somBiPOLES group spent less time in the center compared to control mCerulean (Two-way ANOVA, time: $F_{(2.58, 33.58)} = 2.792$, $p = 0.06$, opsin: $F_{(1,13)} = 5.632$, $*p = 0.03$, time x opsin interaction: $F_{(5,65)} = 1.132$, $p = 0.35$, somBiPOLES $n = 7$ mice, mCerulean $n = 8$ mice). **f** Locomotion in open field test, as distance travelled in the arena in meters, epochs are averaged. (Two-way ANOVA, time: $F_{(1.73, 22.47)} = 19.63$, $*p < 0.001$, opsin: $F_{(1,13)} = 0.454$, $p = 0.51$, time x opsin interaction: $F_{(2,26)} = 0.003$, $p = 0.997$, somBiPOLES $n = 7$ mice, mCerulean $n = 8$ mice). **g** Strategy for recording neuronal activity from aIC-BLA neurons in wild-type mice. AAV9-DIO-GCaMP6m was injected in aIC and CAV2-Cre in BLA and an optical fiber was implanted into aIC. **h** Representative images of GCaMP6m expression in aIC neurons projecting to BLA. **i** Fiber photometry signal recorded from aIC-BLA neurons, (Top) Bulk GCaMP6m signal, and (Bottom) filtered GCaMP6m signal for calcium transient detection. ΔF/F represents the fluorescent changes from the mean level of the entire recording time series. **j** Representation of

automated transient detection. Filtered GCaMP6m peaks exceeding the threshold (horizontal line in the lower trace) were identified as transients. **k** Averaged heat map of global calcium signal recorded from aIC-BLA neurons during EPM test. **l** Global calcium signal is increased in the open arms compared to the closed arms (Two-tailed paired $t$-test, $t = 3.174$, $**p = 0.005$, $n = 20$ mice). **m** Calcium transients frequency is increased in open arms compared to closed arms (Two-tailed paired $t$-test, $t = 3.249$, $**p = 0.004$, $n = 20$ mice). **n.** Averaged heat map of global calcium signal recorded from aIC-BLA neurons during OFT test ($n = 18$ mice). **o** Global calcium signal is increased in the center compared to the border of the OFT (Two-tailed paired $t$-test, $t = 2.249$, $*p = 0.021$, $n = 18$ mice). **p** Calcium transient frequency tends to increase in the center compared to the border of the OFT (Two-tailed paired $t$-test, $t = 2.091$, $p = 0.052$, $n = 18$ mice). **q** Average calcium signal along the position in the open and closed arms of the EPM. **r** Calcium signal in the open and closed arms increases at the extremity (25-35 cm) compared to the beginning (0–10 cm) of the arms ($n = 20$ mice, Two-way ANOVA, $F_{(1,19)} = 6.301$, $*p = 0.02$ for the arms (Open, Closed), $F_{(1,19)} = 6.8$, $*p = 0.017$ for the position (0–10 vs 25–35 cm) and $F_{(1,19)} = 4.509$, $*p = 0.047$ for arms x position interaction; Bonferroni test for open arms $***p < 0.0001$ and for closed arms $**p = 0.006$), and the signal is higher at the extremity of the open than the closed arms (Bonferroni test $***p < 0.0001$). **s** Average calcium signal along the position in the open arms of the EPM while the animal is navigating OUT to the end of the arms or BACK to the center. **t** Average signal is higher when mice are at the extremity compared to the beginning of the open arm only for the OUT direction ($n = 18$ mice, Two-way ANOVA, $F_{(1,17)} = 8.42$, $**p = 0.01$ for the position (0–10 vs 25–35 cm), no effect of direction (OUT, BACK), and $F_{(1,17)} = 7.95$, $*p = 0.012$ for position x direction interaction; Bonferroni test for OUT $**p = 0.0015$). **u** Differential calcium transients frequency (open-closed) correlates with the time mice spent in open arms (One-tailed Pearson correlation: $R^2 = 0.3288$, $**p = 0.0041$, $n = 20$ mice). All the results are represented as mean ± SEM.

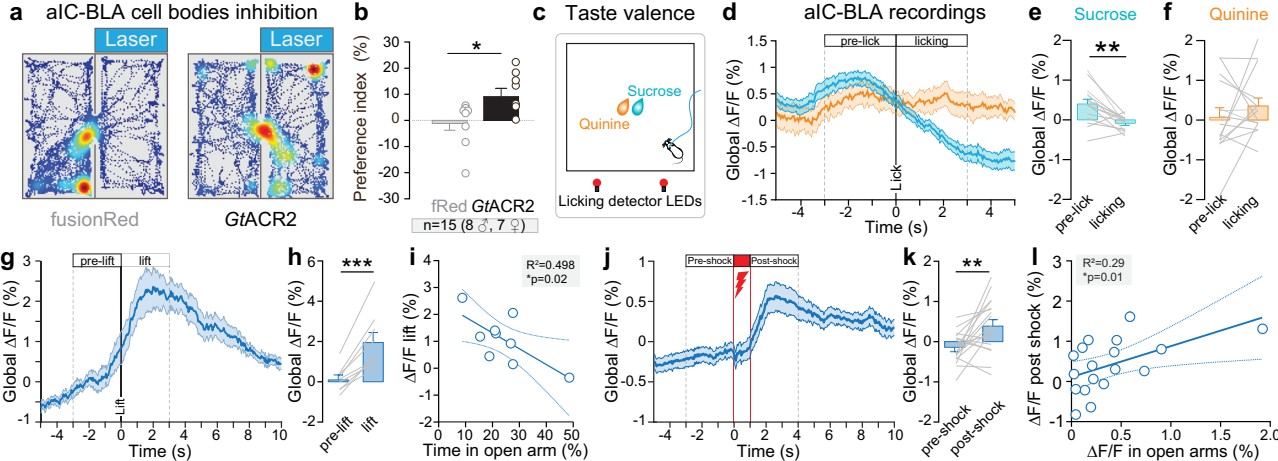

**Fig. 6 | Role of aIC-BLA neurons in valence-related behaviors. a** Two representative traces showing an occupancy heatmap of the time spent in the non-stimulated (left) or stimulated (right) side for a control eYFP mouse (left) and GtACR2 mouse (right). **b** Preference index is increased in *Gt*ACR2 group where aIC-BLA glutamatergic neurons are inhibited compared to fRed control group (Two-tailed unpaired $t$-test, $t = 2.345$, $*p = 0.035$, fRed $n = 8$ mice, *Gt*ACR2 $n = 7$ mice). **c** Schematic of sucrose or quinine consumption test. **d** Peri-licking analysis of the calcium signal between sucrose (blue) or quinine (orange) pre-lick and after licking onset. **e** Global calcium signal after sucrose licking onset is decreased compared to pre-lick (Two-tailed paired $t$-test, $t = 4.077$, $**p = 0.001$, $n = 15$ mice). **f** Bar plot of global calcium signal during quinine pre-lick and after licking onset ($n = 15$ mice).

**g** Peri-event analysis of the calcium signal between pre-suspension and suspension. **h** Global calcium signal is increased during the suspension compared to pre-tail suspension (Two-tailed paired $t$-test, $t = 6.196$, $***p = 0.0003$, $n = 9$ mice). **i** The intensity of calcium signal during tail suspension is negatively correlated with the time mice spent in the open arms of the EPM (One-tailed Pearson correlation: $R^2 = 0.498$, $*p = 0.02$, $n = 9$ mice). **j** Peri-event analysis of the calcium signal between pre- and post-shock ($n = 18$ mice). **k** Global calcium signal is increased during post-shock compared to pre-shock (Two-tailed paired $t$-test, $t = 3.281$, $**p = 0.0044$, $n = 18$ mice). **l** Global signal post-shock correlates positively with global signal in the open arms of the EPM (One-tailed Pearson correlation: $R^2 = 0.2846$, $*p = 0.01$, $n = 18$ mice). All the results are represented as mean ± SEM.

Altogether, when comparing the neural response of aIC and pIC during anxiety-related behaviors, we demonstrated an antero-posterior dichotomy of the insula, where aIC glutamatergic neurons represent anxiogenic spaces, and is linked to the mice level of trait anxiety, suggesting aIC excitatory neurons have an anxiogenic function.

Excitatory neurons of the aIC and pIC have also been shown to differentially control emotional valence, especially for tastants[14]. Using

fiber photometry recordings of excitatory neurons, we found an increase of pIC excitatory neurons activity during sucrose and quinine consumption, whereas aIC excitatory neurons activity was specifically increased during sucrose consumption (Fig. 2a-c). These results are consistent with studies describing the aIC as selectively coding for sweet tastants[31], but contrast with the studies reporting the pIC as selectively coding of bitter tastant[31]. Overall, our data seem to be closer

to the reports describing spatially distributed representation of tastants within the anterior and posterior insular cortex[40].

Notably, in our recordings, both aIC and pIC activity started to rise before sucrose licking onset. Thus, we analyzed the calcium signal of aIC and pIC at the initiation of the movement towards the sucrose port and observed an increase of aIC neurons activity, time-locked to the movement onset. These results suggest that aIC neuronal activity also encodes an anticipatory factor of reward, potentially triggered by the sound of the solution dispenser and/or the smell of sucrose.

Additionally, we showed that aIC glutamatergic neurons activity was increased during tail suspension and after footshocks, whereas these stimuli did not change the activity of pIC glutamatergic neurons (Fig. 2h-m). Finally, we found that aIC global calcium signal post-shock is negatively correlated with trait anxiety levels, whereas a positive correlation is observed within the pIC (Fig. 2n, o). Altogether, this supports the existence of a functional antero-posterior dichotomy, where the anterior insula is selectively encoding non-gustatory negative stimuli.

### Long-range architecture of the insula-amygdala circuit

Seminal and modern studies highlighted the amygdala as a downstream target of the insula[24,26,27,41,42]. Among the twelve regions of interest we assessed with anterograde mapping, we found that the BLA receives the strongest projection from the aIC, while the CeL receives the strongest inputs from the pIC (Fig. 3a–e, Supplementary Fig. 4a–g). However, our data also confirm the existence of input from the aIC to the CeA[43,44], and reveal the existence of pIC-BLA projections, challenging the existence of two segregated insula-amygdala pathways (aIC-BLA and pIC-CeA)[27]. Additionally, we demonstrated that aIC-BLA neurons have dense collaterals, especially in the ipsilateral CeL and NAcore, and controlateral BLA and aIC, which suggests that aIC-BLA neurons recruit a larger network that the BLA.

As the BLA is the main input nucleus of the amygdala, while the CeM is the main output of the amygdala, we characterized inputs from the insula onto those two regions, hypothesizing they could bear different electrophysiological properties. We validated the existence of insula-BLA and insula-CeM functional and monosynaptic excitatory synapses, along with recruitment of polysynaptic inhibition. Since the BLA is mainly composed of excitatory neurons, the polysynaptic inhibition might be carried by feedforward and/or feedback inhibition. On the other hand, as most CeM neurons are inhibitory, the polysynaptic inhibition in this region is likely supported by feedforward inhibition. Interestingly, we observed short-term synaptic depression in both insula-BLA and insula-CeM synapses, with a stronger depression in insula-BLA than insula-CeM synapses. Finally, we identified different intrinsic properties of insula-BLA and insula-CeM neurons, also supporting a model where these two pathways could underlie divergent functions.

### Anxiety-coding properties and anxiogenic function of aIC-BLA neurons

Neural recordings showed that the calcium signal in aIC-BLA neurons is increased in the anxiogenic zones of the EPM and OFT. Moreover, we showed that the activity increases at the extremity of the open arms of the EPM, selectively when mice are travelling OUT into the open arms. As the end of the open arms is more exposed, it is more anxiogenic than the beginning of the arms. Thus, our results suggest that aIC-BLA neurons are proportionally activated with the anxiogenic property of space. Interestingly, we found that in aIC-BLA neurons, the difference of transients between open and closed arms is positively correlated with the anxiety level of the animals, estimated by the time spent in the open arms of the EPM, which support the involvement of this neuronal population in trait anxiety. To note, the difference of transients in center-border of the OFT did not correlate with the time spent of the center. This difference with the correlation observed for the EPM

might be due to the fact that the center of the OFT is less anxiogenic than the open arms as supported by higher time spent in the center of OFT than the open arms of the EPM.

Thus, to characterize the causal role of aIC-BLA neurons in anxiogenic response, we used an optogenetic tool for both neuronal excitation and inhibition (somBiPOLES)[35] and revealed an anxiogenic effect of aIC-BLA neurons manipulation, without real-time effects of neural activation or inhibition in the EPM and OFT. Indeed, the averaged time spent in the center of the OFT is decreased by the alteration of aIC-BLA neural activity, independently of the manipulation type (activation, inhibition, no manipulation) and the same trend was observed in the EPM. The difference of significance between the EPM and OFT is potentially due to the fact that the OFT is a less anxiogenic environment than the EPM, and it is therefore more likely to increase anxiety-related behavior in this environment. Although, these results could appear inconsistent with aIC-BLA anxiety-coding properties, they also suggest that activation of aIC-BLA neurons triggers activation of a large downstream network, in part through collaterals to CeL and NAc (Fig. 3k), leading to a lasting brain state change and therefore increasing anxiety-related behaviors (Supplementary Fig. 8k). The implication of aIC-BLA neurons in the control of anxiety-related behaviors and their increased activity in anxiogenic spaces suggest that aIC-BLA neurons real-time activity is encoding long-term behavioral outcomes on anxiety. This hypothesis is consistent with a model proposing that insular neurons code information at multiple time scales[45,46]. Together, these results show that aIC-BLA neurons control anxiety-related behaviors and encode anxiogenic spaces.

### Valence-coding properties of aIC-BLA neurons

To assess the causal role of aIC-BLA neurons in emotional valence, we used optogenetic manipulation in a real-time place preference/aversion assay. Interestingly, inhibition of aIC-BLA neurons drives place preference, while activation only induces a trend for a preference. Whereas these results could appear conflicting, as both activation and inhibition of the same neuronal population would induce a similar behavioral outcome, this could also imply that activation and inhibition of aIC-BLA neurons both alter the coding properties of the manipulated neurons, leading to similar effects. Thus, to disentangle how aIC-BLA neurons encode valence-related behavior, we recorded their neuronal activity using fiber photometry. Our data show bidirectional coding properties of aIC-BLA neurons which are inhibited by stimuli of positive valence and excited by different stimulations of negative valence. Indeed, while calcium signal in aIC-BLA was increased after footshocks and during tail suspension, the activity of the same neurons was not changed by quinine consumption or during the footshock itself. It suggests that aIC-BLA glutamatergic neurons contribute to coding negative valence stimuli but not aversive tastants or pain.

### aIC-BLA neurons as a neurobiological link between valence and anxiety

Our study brings collective evidence on the role of aIC and aIC-BLA neurons activity in coding both valence and anxiety-related behaviors. Indeed, aIC neural activity when mice are located in anxiogenic spaces is strongly and negatively correlated with the time they spent in the center of the OFT (Fig. 1l, n). This observation is positively linking the activity of aIC during state anxiety, with the level of trait anxiety (Supplementary Fig. 8l). Moreover, calcium signal of aIC excitatory neurons during post-shock is highly and negatively correlated with the time spent in the open arms (Fig. 2o), which positively links aIC neural activity during negative valence with trait anxiety (Supplementary Fig. 8l). As this correlation is mainly observed in response to footshock, which has the strongest negative valence across the tests we performed, it indicates that the link between valence and anxiety-related behaviors might be influenced by the intensity of the stimulus.

Moreover, aIC-BLA differential activity between open and closed arms, and activity during tail suspension are negatively correlated with the time mice spent in the EPM open arms, linking the activity of aIC-BLA neurons with both trait anxiety and negative valence (Supplementary Fig. 8l). Finally, calcium signal of aIC-BLA neurons in the open arms is positively correlated with the calcium signal of the same group of neurons right after a footshock, demonstrating that aIC-BLA neurons are proportionally activated by anxiogenic spaces and a stimulation of negative valence.

Altogether, aIC neurons activity in anxiogenic spaces and in response to a stimulus of negative valence is proportional to trait anxiety level, pinpointing the aIC as a common substrate for both state and trait anxiety as well as negative valence. Interestingly, among aIC excitatory cells, aIC-BLA neurons appear as a crucial building block of the neural circuitry linking anxiety- and valence-related behaviors.

## Outlook

Clinical and preclinical studies report both the insula and amygdala as key brain regions involved in several psychiatric disorders, including anxiety disorders and addictions, which are both characterized by disruption of valence assessment[11,13,43,47,48]. Patients with anxiety disorders present an attentional bias for stimuli of negative valence[7], while in drug addiction, the attribution of positive valence to drug-related cues participates in relapse in humans and animal models[49–51]. Thus, our findings provide a starting point to further characterize the role of insula-amygdala circuits in psychiatric disorders including addiction and anxiety disorders.

## Methods

### Experimental animals and housing conditions

All the experiments were performed with adult male and female C57Bl6/J mice, aged 9–10 weeks at the beginning of the experiments from Charles River Company. The animals were housed in controlled conditions of temperature (range, 20–22 °C) and humidity (range, 50–55%) environment, on a reversed 12 h light/dark cycle and with *ad libitum* access to food and water. Animal maintenance, treatments, and experimental procedures were conducted according to French governmental regulations, and approved by the ethical committee and the Ministry of Education, Research and Innovation (Saisine #16579) in accordance with the guidelines of the European Communities Council Directives.

### Stereotaxic surgeries

Before the surgery, mice were injected with Metacam (5 mg/kg). Mice were anesthetized with isoflurane (5% induction, 1.5–2.0% maintenance) in a stereotaxic frame (Kopf) throughout the entire surgery. During the whole surgical procedures, animals' eyes were protected with Ocry-gel (Laboratoire TVM), and the body temperature was kept at physiological levels, by heating pads.

**Injections.** Intracranial injections were performed using glass pipettes (3-000-203-G/X, Drummond) made by a puller (PC-100, Narishige) to deliver the retrograde tracer or viral vectors at a rate of 1-5 nL/s using a Nanoject III (3-000-207, Drummond). After completion of the injection, the pipette was raised 100 μm, left for additional 10 min to allow diffusion of the retrograde tracer or the viral vector at the injection site, and then slowly withdrawn. After surgery, the mouse body temperature was maintained using a heat lamp until the animal fully recovered from anesthesia.

**Optic fiber implantation.** For fiber photometry and optogenetic experiments, the optic fiber was implanted after the viral vector injections. An optic fiber (0.39 numerical aperture (NA), photometry: 400 μm diameter, optogenetics: 300 μm diameter, Thorlabs) was inserted in a metal ferrule (1.25 mm), polished to obtain 90% and 80% efficiency, and implanted 50 μm or 400 μm above the viral vector injection site for photometry and optogenetic experiments, respectively. Biocompatible cement and resin were used to keep the implanted fiber fixed in the brain. At the end of the surgery, the incision was sutured, and the mouse body temperature was maintained using a heating lamp until the animals are fully recovered from anesthesia. Mice returned to their home cages for 3 to 5 weeks to allow viral expression. For details about the coordinates and viral vectors used for each experiment see Supplementary Table 1.

**Surgeries for fiber photometry recordings.** For region-specific fiber photometry, an adeno-associated virus of serotype 9 (AAV9) carrying the GCaMP6f gene, under the control of the CaMKIIα promoter (AAV9-CaMKIIα-GCaMP6f-WPRE-SV40) to target glutamatergic neurons was injected into aIC or pIC, and an optical fiber (400 μm core, 0.39 NA, >90% efficiency) was implanted 50 μm above the injection sites. For projection-specific fiber photometry, a dual virus strategy was used with an AAV9 carrying the GCaMP6m gene, under the control of the human synapsin promotor (hSyn), for exclusive neuronal expression, and in a double-floxed inverted open reading frame (DIO), to be expressed in a cre-dependent manner injected in the aIC (AAV9-syn-Flex-GCaMP6m-WPRE-SV40). The retrograde vector CAV2 vector expressing the recombinase Cre was injected in the BLA (CAV2-Cre).

**Surgeries for pharmacological experiments.** Stainless steel internal guide cannulae (4 mm long, 26GA, Phymep) were bilaterally implanted in the aIC. Guide cannulae were cemented as described for fiber photometry implants at the following coordinates from Bregma in mm: 1.7 anterior, ±3.1 lateral, and 2.5 ventral. After the cement dried, caps were screwed in the guide cannulae (small cap, Phymep).

**Surgeries for optogenetic manipulations.** [1] For somBiPOLES experiments, an AAV9 carrying the soma-targeted BiPOLES (somBiPOLES) gene under the control of the human synapsin promotor (hSyn) and in a double-floxed inverted open reading frame, (AAV9-hSyn-DIO-somBipoles-mCerulean, donated from Simon Wiegert, University of Hamburg) was injected bilaterally in the aIC. AAV9-CAG-Flex-mCerulean (donated from Simon Wiegert, University of Hamburg) was injected bilaterally in aIC, as control. In the BLA, CAV2-Cre or AAVrg-hSyn-Cre were injected bilaterally and counterbalanced between the mice injected with the somBiPOLES or the control virus. SomBIPOLES is a soma-targeted membrane-trafficking optimized variant generated by fusing an additional trafficking signal from the potassium channel Kv2.1 to the C-terminus of Chrimson. The addition of the soma-targeting signal from the mammalian potassium channel Kv2.1 yielded somBiPOLES, leading to further enhancement of trafficking to the plasma membrane at the soma and proximal dendrites and lower expression in the axons (Supplementary Data Fig. 5a)[35]. [2] For optogenetic excitation, an AAV5 carrying the Channelrhodopsin 2 (ChR2) gene under the control of elongation factor 1a (EF1a) and in a double-floxed inverted open reading frame (AAV5-Ef1a-DIO-hChR2(E123T/T159C)-EYFP) was injected in the aIC, and CAV2-Cre was injected in the BLA unilaterally. For control mice, an AAV5-EF1a-driven vector expressing eYFP in a cre-dependent manner was injected unilaterally in aIC and CAV2-Cre in BLA. [3] For optogenetic inhibition, a synapsin-driven AAV1 carrying the gene coding for the soma-targeted anion conducting modified-Channelrhodopsin 2 (st*Gt*ACR2), in a single-inverted floxed open reading frame (SIO) fused to FusionRed (AAV1-hSyn1-SIO-st*Gt*ACR2-FusionRed) was injected bilaterally in the aIC, and CAV2-Cre was injected bilaterally in the BLA. For control mice, an AAV1, hSyn-driven, expressing mCherry in cre-dependent manner was injected bilaterally in aIC and CAV2-Cre in BLA (AAV1-hSyn-DIO-mCherry).

**Surgeries for anatomical characterization of IC downstream targets.** To label the projections, an anterograde viral vector carrying the gene coding for a green fluorescent protein (AAV9-CaMKIIα-ChR2-eYFP) has been injected in the aIC or in the pIC (Fig. 3a, Supplementary Fig. 2). To label aIC-BLA collaterals, we used a mix of two cre-dependent adeno-associated viral vectors injected in the aIC containing 1/3 by volume of the serotype 8/2 vector expressing an enhanced yellow fluorescent protein (AAV8/2-DIO-eYFP) and 2/3 by volume of a serotype 9 vector expressing Synaptophysin fused to a red fluorescent reporter (AAV9-DIO-Synaptophysin-mCherry). The retrograde canine adenovirus CAV2-Cre was injected in the BLA.

**Surgeries for ex vivo electrophysiology.** To measure the dynamic properties of the IC-BLA or IC-CeM synapses, AAV9/2-CaMKIIα-hChR2(E123A)-eYFP-WPRE was injected in the IC (Fig. 4a). hChR2(E123A) is the hChR2 variant CheTA 2.0 for ultrafast optogenetic control[52,53]. To analyze the intrinsic property of the insular neurons projecting to the BLA and CeM, CTB coupled to AF555 or AF647 (Fischer Scientific) were injected in the BLA or CeM, respectively, to label the insular neurons projecting to the BLA or CeM (Fig. 4h–j).

## Behavioral assays

One week before the fiber photometry and optogenetic experiments, the animals were handled for 30 min a day, at least once a day, and habituated to being connected to the optic fiber. All tests are performed during the dark phase of the reversed light/dark cycle, using red light to observe the behavior ($15 \pm 3$ lux near the behavioral set-ups). The behavioral mazes were always cleaned with acetic acid 2%, and distilled water, before each animal session. The animal weight was measured every day, before the test. One camera on top of the experimental maze was used for all the tests, to track the animal's position. Video recordings were synchronized with the photometry signal recordings. An additional camera was put on the side of the apparatus for tail suspension test. The temperature and humidity of the experiment room were controlled at 20 to 24 °C and 45% to 65%, respectively.

**Anxiety assays.** We used two classical anxiety tests, the elevated plus maze (EPM) and the open field test (OFT). In these tests, the apparatus is divided into an anxiogenic area and a non-anxiogenic area.

- *Elevated plus maze*: The arena has a plus shape ($75 \times 75$ cm) at 60 cm height, consisting of two open arms (anxiogenic space) and two closed arms (non-anxiogenic space) of 5 cm width. Each animal was placed at the entry of the open arm facing the center of the maze and left to explore it for 15 minutes.
- *Open field test*: In this test, the arena is squared ($60 \times 60$ cm), and the anxiogenic space is the center, while the non-anxiogenic space are the borders the field. The center is defined as 50% of the total OFT arena. Each animal was put in the open field and left to explore it for 15 or 20 min.

**Valence assays.**

1. *Sucrose/Quinine test:* quinine is a bitter substance aversive to mice. As mice will taste it only once, we developed a device that will present either sucrose or quinine, in order to entice the animals to consume quinine multiple times. Sucrose (15% sucrose in tap water) or quinine (1 mM) were available in the center of a $60 \times 60$ cm squared arena with transparent walls, following 20 h of food deprivation (for food deprivation, the litter was always changed to remove food scrubs). Sucrose and quinine solutions were respectively presented 17 and 13 times for 20 s each in a pseudo randomize order. Sucrose and quinine consumption were measured using 2 LEDs that respectively blinked each time the animal licked in the sucrose or quinine port.

2. *Mild footshock*: in order to assess the neuronal response to another negative valence stimulus, we applied 10-15 mild footshocks at 300 µA for 1 s every minute.

3. *Tail suspension*: the tail suspension test (TST) was used as an aversive experience. The mice were suspended 3 times by the hand of the experimenter at approximately 40 cm above their home cage for 1 min every minute.

**Real-time place preference/aversion (RTPP/A).** This assay was performed over two days for 20 min session each day, in a squared area ($60 \times 60$ cm) divided into two equal chambers. Mice have free access to both chambers through an opening in the central separation. To differentiate the chambers, one was assigned alternating black and white vertical stripes on its walls, while the other with a black circle on the wall. For light optogenetic stimulation protocols see optogenetics section.

## Fiber photometry recordings

**Signal recordings.** To record the fluorescence signals, the fiber photometry system was custom-made and configured as previously described[54,55]. The optical fiber implanted in the aIC or pIC transmitted the light emitted by the LEDs to excite GCaMP6 chromophore. The 20x objective was connected to a complementary metal-oxide semiconductor (CMOS) camera that detected the fluorescence level, which is indicative of the calcium levels and neural activity in vivo. The optimal wavelength for GCaMP6 excitation is 470 nm, while the isosbestic wavelength is 405 nm (Supplementary Fig.1a). This 405 nm channel was used as a control, since it is independent of calcium concentration, and allowed us to remove motion-related artifacts and signal unrelated to neuronal activity. Two excitation LEDs (470 nm and 405 nm, Thorlabs) were bandpass filtered and used to excite calcium-dependent and -independent fluorescence from GCaMP6. The filtered excitation lights were reflected by a 495 nm long-pass dichroic mirror and then delivered to the target brain region through a patch cord (0.29 NA, 400 µm) coupled with the implanted fiber. To minimize the photobleaching effect of the recording, the light intensities in the tip of the patch cord were adjusted to ~140 µW for the 470 nm channel and ~58 µW for the 405 nm channel. The GCaMP6 emission light was bandpass filtered and focused on CMOS camera sensor for fluorescence detection. A custom Matlab script was used to synchronize fiber photometry and video recordings (revised from: https://github.com/deisseroth-lab/multifiber), combined with a programmed Arduino UNO board. The sampling rate was settled at 20 Hz for both photometry and video recording.

**Data analysis.** After the behavioral tests, photometry recordings were analyzed using custom Matlab scripts achieving the following functions:

1. Remove the first minute of the recording to avoid LED stabilization and photobleaching artefacts. Each recorded fluorescence image was synchronized with the GCaMP6 light excitation made by the LEDs (470 nm and 405 nm).

2. Normalize the 470 nm and 405 nm signals over the test, by subtracting the mean fluorescence of each channel from the fluorescence recorded at each time point and dividing this value by the mean fluorescence (($F$-$F_{mean}$)/$F_{mean}$) = $\Delta F/F$). The mean fluorescence was calculated over a 60 s sliding window, in order to remove photobleaching effects occurring over time.

3. Fit the normalized 405 nm signal to the 470 nm signal using a linear fit to standardize the mean and the amplitude of the 405 nm signal. When GCaMP6 is excited by 405 nm wavelength, the amount of fluorescence emitted is independent of the calcium concentration, thus this signal only carries non-neuronal changes including potential movement artifacts. On the contrary, when GCaMP6 is excited with 470 nm wavelength, the recorded signal

contains physiological data and potential artifacts. The goal of this first analysis steps is to normalize the calcium-independent 405 nm signal to the 470 nm signal in order to subtract the fitted 405 nm signal from the 470 nm signal to remove potential movement artifacts.

4. Subtract the calcium independent signal (fitted 405 nm) from the normalized GCaMP6 signal (470 nm), to eliminate unspecific fluorescence (including potential movement artifacts): $\Delta F/F = \Delta F/F_{470} - \Delta F/F_{fitted405}$. The obtained result was the global signal ($\Delta F/F$) used as an estimate of tonic activity of the recorded neurons, and represented in %. For each sample, 100% would represent $\Delta F/F$ being twice the value of the average F, calculated over 60 s around the sample.

5. Detect calcium transients: the global signal was bandpass filtered (low threshold 0.2 Hz, and high threshold 6 Hz). Filtered peaks were detected as high-amplitude events (defined as events with amplitudes two median absolute deviation (MAD) above the median of the sliding window), were then filtered out from the signal, and the median of the trace re-calculated excluding those peaks. Peaks with a local maxima greater than two MADs of the resultant trace, are identified as transients, which is used as an estimate of phasic activity of the recorded neurons. Average peak amplitude and peak frequency were compared across groups (adapted from Muir et al.)[55].

6. Synchronization of the fluorescence signal with the location of the animal in the maze (EPM or OFT), recorded by the behavior camera and detected using the Bonsai software, or with the discrete events (sucrose/quinine licking bout onset, footshock, and tail suspension). For sucrose and quinine consumption, each licks were detected thanks to an infra-red beam break, and a bout of licks was defined as one or several licks separated by a minimum of 10 s interval.

7. Map the intensity of the global calcium signal depending on the location of the animal in the entire mazes (heatmaps EPM or OFT, Figs. 1 and 5): video frame was synchronized with the respective photometry sample. Animal's position within the apparatus which was subdivided in pixels was extracted using Bonsai software allowing to obtain the instantaneous coordinates of the mice for the entire duration of the test. Based on the dimension of the apparatus, the coordinates of the mice were transformed from pixels to centimeters into 0.5 cm spatial bins. The Occupancy Map was built based on the number of frames spent in each spatial bin. The Cumulative Signal Map was constructed by summing the activity in each spatial bin according to the animal position frame by frame. Finally, the cumulative map was divided by the occupancy map to obtain the average signal map.

8. Calcium signal analysis based on the mice position into the arms and the travelled direction: transform video-tracking data form pixels were transformed to cm and binned in 0.5 cm to calculate the averaged signal in each bin by dividing the sum of the signal by the number of visits of each bin during the analyzed time period. Then the trajectory was linearized by averaging the bin values across the smallest dimension of each arm in order to analyze the position within each EPM arm.

9. To compare trajectories in the EPM by clustering them into two categories OUT (going out from the center of the EPM toward the end of the open arms) and BACK (coming back to the center), only discrete movements were analyzed by applying a minimum threshold of velocity (3.75 mm/frame). This threshold was established by observing the histogram of the velocities in the EPM, and reflect movement.

10. The global signal was averaged depending on the location of the animal in the maze (open arm, closed arms and center of the EPM, or center and borders of the OFT) or around the discrete events (before after licking, shock or lift). These data are represented in the bar graphs in Fig. 1c–f and Fig. 5l, m, o, p.

In the end, we obtain three variables as results: the global signal, transients frequency, and the transients amplitude.

## Pharmacology experiments

**Behavior**. Three days after surgery, mice were handled for two days to habituate them to be restrained and to have connections to their guide cannulae. On the third day, mice were brought to the behavioral room for one hour, before inserting an internal cannula (5 mm long, 33GA, Phymep) into the two implanted 26GA guide cannulae. Coordinates from Bregma of the tip of the internal cannulae in mm: 1.7 anterior, ±3.1 lateral, and 3.5 ventral. The internal cannulae were connected to a Hamilton syringe place into a pump (KDS legato 101, Phymep), and loaded for half of the mice with the 5-HT1A agonist 8-OH-DPAT, or with the vehicle for the other half of the mice. A volume of 150 nL, containing 200 ng of 8-OH-DPAT, or 150 nL of vehicle was infused into each aIC at a rate of 2 nL/s (1 min 15 s of injection). The internal cannulae were withdrawn 2 min after the end of the infusion, and mice were placed in the center of the EPM, facing an open arm, 2 min after withdrawal of the internal cannulae. After 15 min of EPM test, mice were removed from the maze, and cannulae caps were placed back in the guide cannulae.

**Drug**. The 5-HT1A agonist R-(+)-8-Hydroxy-DPAT hydrobromide (8-OH-DPAT, Tocris) was diluted in Ringer to obtain a solution at 1.08 mM of 8-OH-DPAT (200 ng/150 nL). The Ringer was containing (in mM): 154.1 $Cl^-$, 147 $Na^+$, 2.7 $K^+$, 1 $Mg^{2+}$, and 1.2 $Ca^{2+}$, adjusted to pH 7.4 with 2 mM sodium phosphate buffer.

**Placement verification**. The day after EPM test, the procedure was repeated by injecting 150 nL of CTB-555 at 2 nL/s (0.4%, Thermo Fisher Scientific) to confirm the placement of the internal cannula (Fig. 1p).

**Data analysis**. The software Bonsai was used to track the mouse location and the data were analyzed with custom Python script to extract the time spent in the open and closed arms of the EPM as well as the distance travelled was also measured for both tests.

## Optogenetics

**somBiPOLES experiments**. Six weeks after surgery, mice were habituated to be handled and to be connected to the optic fiber. Videos were recorded using a webcam (Logitech). Bilateral stimulation of aIC neurons was achieved by connecting the fiber implant to a 1 × 2 Step-index multimode fiber optic coupler (200 μm diameter, 0.39 NA, Thorlabs, Germany) in turn connected to a 1 × 2 Fiber-optic Rotary Joints–Intensity Division (RJ-ID, 400–700 nm, Doric Lenses, Canada) 200 μm diameter). The rotary joint splitter was fixed on top of the arena and connected on the other side through a patch cord to a mini-cube wavelength divider (DMC_1×2w_FC, [470 nm:590 nm,470], Doric Lenses, Canada) which functions as a laser combiner system housing patch cords from a 473 nm laser source (MBL-III-473-100 mW, Opto-Engine, USA) and a 593 nm diode laser (MGL-F-593.5–50 mW, Opto-Engine, USA) for activation of the $Gt$ACR2 and Chrimson components of somBiPOLES, respectively. Coupling to the implant was done with zirconia mating sleeves (1.25, Doric lenses, Canada). Stimuli were generated according to regions of interest, the time epochs defined in custom-written code in Bonsai software and laser was triggered using Arduino-Uno board.

For activation of Chrimson, pulse trains (593 nm, -10 mW at each fiber end, 20 ms pulse duration, 20 Hz repetition rate) of 3 min epoch were presented, while $Gt$ACR2 was activated by pulse trains (473 nm, -20 mW at each fiber end, 5 ms pulse duration, 20 Hz repetition rate). Light emitted was tested to ensure an inter-train interval of 2 s between

each epoch and to not overlap with laser lights, through a custom-written Arduino code. For EPM, the experiment was performed with 3 min epochs, in which one epoch of orange light stimulation was followed by an epoch of blue light stimulation and an epoch of no stimulation. This was repeated once, making a total duration of 18 min. For OFT, the same epochs design was used, but for 4 min each, a total duration of 24 min.

**ChR2 and GtACR2 real-time place preference/aversion (RTPP/A).** On the first day, mice were allowed to freely explore the two chambers with no light stimulation. The natural preference of the animal for one side is characterized by the side where the mice spend more time and is defined as the preferred chamber.

On the second day, according to the preference exhibited, half of the mice were stimulated on their preferred side, and the other half on the non-preferred side. At the start of the 20 min session, individual mice were placed in the unstimulated chamber. For the optogenetic inhibition (GtACR2), each time the mouse entered the stimulated side of the arena, light stimulation (5 ms pulses, 20 Hz, 20 mW, 473 nm laser) was delivered until the mouse crossed back into the non-stimulated chamber. For optogenetic activation (ChR2) each time the mouse entered the stimulated chamber, light stimulation (20 ms pulses, 20 Hz, 10 mW, 473 nm laser) was delivered until the mouse crossed back into the non-stimulated chamber.

The Place preference index was calculated as below: (T time)

$$\text{Place preference index}(PPI)\% = \frac{T_{\text{stim chamber}} - T_{\text{non-stim chamber}}}{T_{\text{stim chamber}} + T_{\text{non-stim chamber}}} \times 100$$

### Brain extraction, histology, and imaging
After fiber photometry recordings, pharmacological experiments, ChR2 or GtACR2 optogenetics experiments, animals were euthanized with pentobarbital (300 mg/kg) and perfused with ringer's solution followed by 4% PFA (antigenfix, F/P0014, MM France) at 4 °C. The brains were extracted and left in 4% PFA overnight at 4 °C, for post fixation. Then, they were transferred to a 30% sucrose solution, in PBS 1× at 4 °C, for cryoprotection. After sinking in the sucrose solution, the brains were embedded in optimum cutting temperature compound (OCT) and sliced into 100-50 μm coronal slices, in a freezing sliding microtome (12062999, Thermo Scientific). Sections were incubated with a DNA-specific fluorescent probe Hoechst33342 (1:5000, Fischer scientific) for 20 min, washed with PBS 1× followed by mounting on microscope slides with Polyvinyl alcohol PVA-DABCO mounting medium (Sigma Aldrich, France). Images were acquired at the fluorescence microscope (THUNDER Imager 3D Tissue, Leica), using the deconvolution system.

### cFos immunohistochemistry
For somBiPOLES optogenetic experiments, cFos stimulation was performed by connected the mice to a patch cord (200 μm diameter, 0.39 NA, Thorlabs, Germany), and orange light (593 nm, ~10 mW at each fiber end, 20 ms pulse duration, 20 Hz repetition rate) was delivered for 5 min, while the mice were in a clean new cage. After an hour and a half, animals were anesthetized with pentobarbital (300 mg/kg) and transcardially perfused. After cryoprotection in PBS 1×-buffered sucrose (30%) solution, brains were sliced to 50 μm coronal brain sections using a frozen sliding microtome (12062999, Thermo-Scientific). Brain sections were first washed in PBS 1× (3 × 5 min) at room temperature (RT) and then were blocked in 3% normal goat serum (NGS) in PBS 1X (0.3% Triton-X-100) for 2 h at RT. Slices were incubated overnight at 4 °C in rabbit polyclonal anti-cFos (1:500, Synaptic System, #226-003). Then, slices were washed with PBS 1X (3 × 10 min), followed by incubation with fluorescent secondary antibodies Alexa Fluor 555 Goat anti-rabbit IgG (1:1000, Fischer scientific, #1890860) at RT for 2 h. After

washing with PBS 1X (3 × 10 min), the slices were incubated with Hoechst33342 (1:5000) for 10 min before a final washing step. Finally, brain slices were mounted, and cover slipped using PVA-DABCO (Sigma Aldrich). Immediate-early gene cFos were used to quantify optogenetic activation in somBiPOLES experiment, as a marker for neuronal activation in aIC-BLA neurons. Images of fiber implantation sites, viral expression, and injection sites were taken using a fluorescence microscope (Leica Microscopy) and were overlaid on the coronal atlas of Paxinos [4th edition][56]. For confocal microscopy, images were captured through a 10x dry objective (NA 0.70) and a 40× oil-immersed objective (NA 1.30) from a Leica SP5 confocal microscope (Leica Microscopy). For cell counting of cFos and insula projections, z-stacks of ROIs (z-step: ~2–4 μm, between stacks, 1024 ×1024 pixels) were scanned using the 40x objective and cell counting was only made on the maximum projection of merged pictures. Leica software and Image-J were used to process images and count cells.

### Inclusion criteria
Mice were excluded from the analysis when optic fiber placement or viral expression patterns were out of brain targets.

### Anatomical characterization of IC downstream targets and aIC-BLA axonal collaterals
Images were acquired with a Leica SP8 confocal microscope (Leica Microscopy). First, all processed brain slices, as well as all slices containing an injection site were imaged using a 10X objective. Z-stacks of the twelve downstream regions have been captured in a picture format of 1024 × 1024 pixels (Number of z-stacks: 30, z-steps: ~ 1 ± 0.10 μm, 2–3 frame average). All images have been processed using the open source Fiji software (ImageJ, NIH). For quantification of eYFP-expressing axonal projections from glutamatergic neurons of aIC and pIC in Fig. 3, the images were then z-projected across 30 slices using a maximum intensity projection. Fluorescence was quantified using custom Python script. Fluorescence values from digitized tiff images in 8 bits encoded as a number between 0 and 255 were divided by 255, to bring the fluorescence intensity range between 0 and 1. A thresholding procedure was used to separate axon fluorescence from background. A threshold of 0.5 was set across all images in the green (eYFP) channel. eYFP fluorescence was quantified as the fraction of pixels in the image above this threshold. For quantification of aIC-BLA collaterals, we use the same protocol. The ipsilateral BLA image served for the normalization of the eYFP and SynP-mCherry fluorescence intensity quantification. A threshold of 0.04 and 0.06 was set across all images in the red (SynP-mCherry) and in the green (eYFP) respectively.

### Ex vivo electrophysiological recordings
**Preparation of acute brain slice.** One week after retrograde tracer injection or 4–5 weeks after viral vector injection, mice were anesthetized with pentobarbital (300 mg/kg) and perfused transcardially, with 20 ml of modified artificial cerebrospinal fluid (ACSF, at ~4 °C) containing (in mM): 75 sucrose, 87 NaCl, 2.5 KCl, 1.3 NaH$_2$PO$_4$, 7 MgCl$_2$, 0.5 CaCl$_2$, 25 NaHCO$_3$, and 5 ascorbic acid. The brain was extracted and 300 μm thick coronal brain slices containing the insula, BLA or CeM were collected by a semi-automatic vibrating blade microtome (VT1200; Leica) inside ice-cold modified ACSF. Acute coronal sections were incubated in oxygenated ACSF containing (in mM): 126 NaCl, 2.5 KCl, 1.25 NaH$_2$PO$_4$, 1.0 MgCl$_2$, 2.4 CaCl$_2$, 26 NaHCO$_3$, 10 glucose (pH 7.25-7.4; 300 ± 2 mOsm). Recordings were started one hour after slicing and the temperature was maintained between 31 and 33 °C both in the holding chamber and during the recordings. All injection sites were checked and imaged with the microscope (BX51, Olympus).

**Whole-cell patch-clamp recording.** Recordings were made from CTB-labelled neurons in the insula for the intrinsic property of

insular-amygdala projection neurons and from neurons in BLA or CeM or CTB-labelled neurons for channelrhodopsin-assisted circuit mapping (CRACM) and their synaptic properties. Borosilicate glass capillaries were pulled by P-1000 puller (Sutter Instrument) and the recording electrodes (4–6 MΩ) were filled with K-gluconate-based internal solution containing (in mM): 125 potassium gluconate, 20 HEPES, 10 NaCl, 3 MgATP, 8 biocytin, and 2 Alexa Fluor 350 (pH 7.25–7.4; 280–290 mOsm). Voltage- and current-clamp recordings were conducted using a Multiclamp 700B amplifier (Molecular Devices). Analog signals were low-pass filtered at 1 kHz and digitized at 10 kHz using a Digidata 1440 and pClamp9 software (Molecular Devices). ACSF and drugs were applied to the slice via a peristaltic pump (Minipuls3, Gilson) at 2 mL/min.

The intrinsic properties (passive and active) of neuronal membrane were measured from all patched neurons. The passive properties of neuronal membranes were analyzed by seal test (5 mV, 1 s) and voltage-step mode ($-110 \pm 30$ mV for 1 s, 10 mV step). In the current clamp recording, the current was injected in ramp mode (0–300 pA for 1 s) for measuring the firing threshold and rheobase and step mode (0–300 pA for 2 s, 20 pA step) for acquiring injected current-action potential frequency relationship.

For CRACM of insular inputs onto BLA or CeM neurons or reciprocal connection between the insula and BLA, ChR2 expressed in the axonal fibers was activated in the downstream regions using a LED light source (470 nm, CoolLED p4000) and the average power to trigger a light response in the patched neurons was $0.17 \pm 0.06$ mW/mm$^2$. To test if the response of ChR2 terminal activation was monosynaptic, optogenetically-evoked excitatory/inhibitory postsynaptic current (oEPSC/oIPSC) was measured with holding potential clamped at -70 mV during the bath application of the sodium channel blocker, tetrodotoxin (TTX, 1 μM) and the potassium channel blocker, 4-aminopyridine (4AP, 100 μM)[17,57]. NMDA-R (glutamate receptor selectively activated by N-methyl-D-aspartate) antagonist, D-(-)-2amino-5-phosphonopentanoate (AP5), and AMPA-R (glutamate receptor selectively activated by 2-amino-3-(3-hydroxy-5-methyl-iso-xazol-4-yl) propanoate) antagonist, 2,3-Dioxo-6-nitro-1,2,3,4-tetra-hydrobenzo[f]quinoxaline-7-sulfonamide (NBQX), were applied to confirm whether the monosynaptic contacts were abolished by blocking all the glutamatergic receptors. At a holding potential of -70 mV for EPSCs and of 0 mV for IPSCs, paired and ten trains of light stimulations (2 ms, 50 ms interval) were given to test short-term synaptic plasticity.

The location of all recorded neurons was checked after the recording (Supplementary Data Fig. 3f) by Allen brain atlas (https://portal.brain-map.org/) and only the cells located in the aIC, pIC, BLA or CeM were kept for further analysis. All the electrophysiological data were analyzed by Clampfit software (Molecular Devices) and custom-made python script.

### Statistical analysis
We used factorial ANOVA and t-tests using Matlab or GraphPad Prism 9. For ANOVA analysis, we followed up on significant main and inter-action effects ($p < 0.05$), with Bonferroni or Dunn's post-hoc test. Because our multifactorial ANOVA yielded multiple main and interaction effects, we only report significant effects that are critical for data interpretation. For correlation analyses, Pearson's correlation coefficient was calculated ($R^2$).

### Reporting summary
Further information on research design is available in the Nature Portfolio Reporting Summary linked to this article.

## Data availability
The data generated in this study are provided in the Source Data file. Source data are provided in this paper.

## Code availability
The custom-made MATLAB script used for behavioral and calcium activity analysis can be downloaded from https://github.com/beyelerlab/FiberPhotometry_anxiety.git.

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

## Acknowledgements

We acknowledge the entire Beyeler Lab and Christian Lüscher for insightful discussions and advice. We thank Aquineuro for their technical support and Aline Desmedt for sharing equipment. We thank the Bordeaux Imaging Center (BIC) for the use of their confocal microscope. We thank Sara Laumond, Julie Tessaire, and the technical staff of the animal housing facility of the Neurocentre Magendie (INSERM U1215) for their invaluable support. We acknowledge the support of the Région Nouvelle-Aquitaine, the 'Avenir program' of the French institute of health (INSERM), the Fondation NRJ-Institut de France, the 'Agence nationale pour la recherche' (ANR), and of the 'Fondation Schlumberger pour l'éducation et la recherche' (FSER) to A.B., as well as the support of the Fondation pour la recherche médicale (FRM) to A.B. and C.N. (ARF201909009147).

## Author contributions

C.N., A.J., Y.W., H.E., S.D., C.F., Y.C., A.M., L.S., A.V., M.M., D.J., and A.B. carried out the experiments and performed data analysis. C.N., A.J., Y.W., S.D., and A.B. designed the study. J.S.W. and S.R.R. provided the viral vectors for the somBiPOLES experiment. C.N., A.J., and A.B. wrote the manuscript. All authors critically reviewed the content and approved the final version before submission.

## Competing interests

The authors declare that they do not have any competing interests or conflicts of interest (financial or non-financial) related to the material presented in this manuscript.
