## [Peer Review File · Nature Communications]

Linking emotional valence and anxiety in a mouse
insula-amygdala circuitEditorial Note: This manuscript has been previously reviewed at another journal that is not operating a transparent peer review scheme. This document only contains reviewer comments and rebuttal letters for versions considered at Nature Communications.

REVIEWER COMMENTS

Reviewer #1 (Remarks to the Author):

Nicolas et al. explored insular cortex (IC) amygdala interactions in encoding and controlling anxiety; using a set of behavioral and electrophysiological studies in mice, they identified a role for anterior (aIC) and the glutamatergic projection from the aIC to the basolateral amygdala (BLA) in encoding aversive valence and anxiety.

To demonstrate this, they first performed bulk Ca²⁺ imaging in the aIC and pIC and related aIC activity to anxiogenic features; chemogenetic silencing of aIC attenuated of anxiety-like behaviors in EPM; bulk Ca signals showed that pIC most sensitive to negatively valenced tastants. The authors then mapped aIC and pIC projections target areas and characterized IC-BLA and IC-medial central amygdala (CeM) projections by optogenetic ex vivo electrophysiology, identifying recurrent monosynaptic IC-BLA connectivity. They then focused on the role of the aIC-BLA projection (as their identified strongest aIC output) and assessed its role in anxiety-like behaviors by projection specific optogenetic manipulations and Ca²⁺- bulk imaging, indicating that aIC-BLA is most sensitive to open but not closed arms in EPM and that optogenetic manipulation modulates anxiety; likewise, aIC-BLA activation modulates place preference and encodes aversive valence; overall this falls in line with their observations on aIC itself.

The strength of this study rests in its accessibility and straightforward focus on highlighting valence and anxiety unifying circuitry as well as in the projection specific dissection and electrophysiological characterizations of the aIC-BLA circuit. As such it represents a concise and important contribution to the field.

However, as I detail below, the manuscript could be improved in the imaging and behavioral domains, as some results are in part incoherent, also in the context of other studies.

As also noted in the previous round of review/by other reviewers, the key issue was additional support for the specific roles aIC-BLA outputs in valence and anxiety in light of their partially inconsistent behavioral and imaging data sets. While the authors have attempted to resolve these issues as much as possible, the authors are strongly encouraged to resolve this by attempting unifying interpretations

and/or graphing a working model. This will also help to communicate the valence/anxiety concept addressed in this study.

MAJOR/Anxiety

The authors are encouraged to rephrase their integration of the imaging and behavioral data in the discussion by merging the sections in lines 390-421, as they did for the valence in Lines 423-435. In this section they should reconcile their behavior and imaging data into a working model how aIC-BLA operates in anxiety (e.g., low/silenced -> anxiolytic, mid/unperturbed -> neutral, high/activated -> anxiolytic vs. low/silenced -> anxiolytic, mid/unperturbed neutral, high/activated -> anxiogenic)*. This is important since the title conclusions are drawn from the partially inconsistent nature of behavioral (Fig. 5d-e) and calcium imaging data sets.

MAJOR/Valence

As both manipulations result in similar behavioral phenotypes (Fig. 6b and Extended Data Fig. 6j, and Figure 5e for that matter), the construct of how activity in this connection represents valence is complex to understand.

The authors are encouraged to therefore rephrase their discussion section Lines 423-453 neurons to better convey their understanding of aIC-BLA valence coding (e.g., low/silenced -> negative, mid/unperturbed -> neutral, high/activated -> positive vs. low/silenced -> positive, mid/unperturbed neutral, high/activated -> negative)*.

*) The manuscript main findings of aIC-BLA in anxiety and valence might be synthesized and sketched into working model in the supplementary material to better communicate the authors working hypothesis for linking valence and anxiety in IC-amygdala circuits.

Minor/Circuit organization

The differentiation of neurons into aIC-BLA and pIC-BLA neurons is highly appreciated. For the reader to keep with the overall flow of the manuscript, the authors may consider moving the more relevant parts of the BLA-projecting insula-resolved data (Extended Data Fig. 3g-i) to the main Fig. 4, and instead shifting the panels comparing between IC-BLA and IC-CeM projectors overall (Fig. 4k,l,m), to the Extended Data Fig. 3. This might help highlighting their key findings.

MINOR/Additional supporting analysis

The authors infer an association of anxiety and negative valence by correlating the activity of aIC-BLA projectors in an anxiogenic space with activity post-shock (Fig.6m). To further back-up this claim I suggest to do the equivalent analysis for positive valence with activity upon sucrose lick and move-to-

port as presented in Fig. 3b,d. Additionally, a negative correlation between sucrose lick/move-to-port activity and shock responses would support the claim that the activity upon shock in aIC-BLA projectors does indeed represent negative valence. The authors are encouraged to explore if such an analysis would lend additional support to their model.

Reviewer #2 (Remarks to the Author):

I have previously favorably reviewed this manuscript for Nature Neuroscience. The last version I already commented on seems identical to the one submitted here and has not been revised in light of 2 reviewers' comments (mine copied below).

As such, in light of my previous comments on this version (not shown in the rebuttal in this submission), regarding "Point #2" from my last comments, I suggest the authors consider more transparency by adding all the non-significant valence-anxiety assays correlations (shown in the rebuttal figures) to the manuscript supplemental figure. This would allow the readers to form a more comprehensive view of the relationship between valence and anxiety in aIC-BLA. After this, I recommend accepting the manuscript for publication.

Here is a summary of my previous comments:

Following additional comments from two reviewers, in this revised version the authors have performed several additional experiments and analyses. We thank the authors for making these efforts to address reviewer concerns. Generally, our previous comments revolved around 2 main issues:

1. Establishing whether aIC, and specifically aIC-BLA, actually have a causal role in valence and anxiety.
2. Establishing a clear link between valence and anxiety in aIC-BLA.

The authors have attempted to address these issues, but we still have some concerns.

Specifically for point #1:

The authors have added 5-HT pharmacology in the aIC as a form of inhibition, which resulted in effects that are consistent with some the main claims of the paper and therefore strengthen the manuscript. However, the results of the optogenetics experiments remain somewhat unclear. First, despite performing ex-vivo controls for BiPOLES, the in-vivo experiments yielded unclear results. The

manipulation seems inconsistent with both inhibition and excitation, and the effect that is statistically significant is weak and only works for one anxiety assay but not for the other. The authors have given some arguments in the rebuttal for why only one assay was affected, but the BiPOLES effects seem more pronounced and statistically significant during the inhibition epochs and these inhibition results are inconsistent with the 5-HT pharmacological inhibition. Therefore, we feel that this BiPOLES experiment still remains inconclusive. Second, inhibition of aIC-BLA seems to have a robust effect in RTPP, but a very similar effect appears to occur with unilateral optogenetic activation of aIC-BLA, although not quite statistically significant ($p=0.07$). Why would both aIC activation and inhibition yield similar effects?

Together, these optogenetics experiments suggest that the causal role of aIC-BLA in valence and anxiety has not been really established in a robust way beyond the 5-HT pharmacology experiment. And it is unclear if this is due to methodological issues or not.

Specifically for point #2:

The authors have indeed added correlations between some of the valence assays (foot shock, tail suspension) and anxiety assays (OFT, EPM). This is a very important and welcome addition. Nevertheless, it is unclear if the relationship between anxiety and valence in aIC and in aIC-BLA is really a robust relationship because most of the correlations are not significant, and only a few are. If this is indeed a general phenomenon, it is unclear why it is apparently inconsistent. Therefore, these experiments also give us an uneasy feeling that the link between valence and anxiety aIC-BLA has not been really established in a robust way.

We would suggest to consider more transparency by adding all the non-significant correlations (shown in the rebuttal figures) to the manuscript. This would allow the readers to form a more comprehensive view of the relationship between valence and anxiety in aIC-BLA.

We would like to greatly thank the reviewers for their positive comments and constructive input, which we believe allowed us to improve our manuscript and its impact on the neuroscientific community. As suggested, we have rephrased and strengthened the discussion with a proposed working model, and included additional correlations between aIC, pIC and aIC-BLA activity with anxiety-related behaviors to increase transparency (Fig. 6i, Extended data Fig. 1(p-q), 2 and 8).

We would also like to thank the reviewers for their positive feedback on our manuscript, including for acknowledging our study's "accessibility and straightforward focus on highlighting valence and anxiety unifying circuitry" and that it "represents a concise and important contribution to the field".

Reviewer #1 (Remarks to the Author)

Nicolas et al. explored insular cortex (IC) amygdala interactions in encoding and controlling anxiety; using a set of behavioral and electrophysiological studies in mice, they identified a role for anterior (aIC) and the glutamatergic projection from the aIC to the basolateral amygdala (BLA) in encoding aversive valence and anxiety.

To demonstrate this, they first performed bulk Ca²⁺ imaging in the aIC and pIC and related aIC activity to anxiogenic features; chemogenetic silencing of aIC attenuated of anxiety-like behaviors in EPM; bulk Ca signals showed that pIC most sensitive to negatively valenced tastants. The authors then mapped aIC and pIC projections target areas and characterized IC-BLA and IC-medial central amygdala (CeM) projections by optogenetic ex vivo electrophysiology, identifying recurrent monosynaptic IC-BLA connectivity. They then focused on the role of the aIC-BLA projection (as their identified strongest aIC output) and assessed its role in anxiety-like behaviors by projection specific optogenetic manipulations and Ca²⁺- bulk imaging, indicating that aIC-BLA is most sensitive to open but not closed arms in EPM and that optogenetic manipulation modulates anxiety; likewise, aIC-BLA activation modulates place preference and encodes aversive valence; overall this falls in line with their observations on aIC itself.

The strength of this study rests in its accessibility and straightforward focus on highlighting valence and anxiety unifying circuitry as well as in the projection specific dissection and electrophysiological characterizations of the aIC-BLA circuit. As such it represents a concise and important contribution to the field.

However, as I detail below, the manuscript could be improved in the imaging and behavioral domains, as some results are in part incoherent, also in the context of other studies.

As also noted in the previous round of review/by other reviewers, the key issue was additional support for the specific roles aIC-BLA outputs in valence and anxiety in light of their partially inconsistent behavioral and imaging data sets. While the authors have attempted to resolve these issues as much as possible, the authors are strongly encouraged to resolve this by attempting unifying interpretations and/or graphing a working model. This will also help to communicate the valence/anxiety concept addressed in this study.

We thank the reviewer for his positive comments and his suggestion to summarize our study by a graphical model now presented in Extended Data Fig. 8j-l. We hope this will clarify how we coherently interpret our imaging and optogenetic results.

MAJOR/Anxiety

The authors are encouraged to rephrase their integration of the imaging and behavioral data in the discussion by merging the sections in lines 390-421, as they did for the valence in

Lines 423-435. In this section they should reconcile their behavior and imaging data into a working model of how aIC-BLA operates in anxiety (e.g., low/silenced -> anxiolytic, mid/unperturbed -> neutral, high/activated -> anxiolytic vs. low/silenced -> anxiolytic, mid/unperturbed neutral, high/activated -> anxiogenic)*. This is important since the title conclusions are drawn from the partially inconsistent nature of behavioral (Fig. 5d-e) and calcium imaging data sets.

We have now fused the sections discussing the optogenetic manipulation and photometry recordings of aIC-BLA neurons regarding anxiety-related behaviors (Lines 398-425), to better convey how we reconcile our imaging and manipulation findings.

MAJOR/Valence

As both manipulations result in similar behavioral phenotypes (Fig. 6b and Extended Data Fig. 6j, and Figure 5e for that matter), the construct of how activity in this connection represents valence is complex to understand. The authors are encouraged to therefore rephrase their discussion section Lines 423-453 neurons to better convey their understanding of aIC-BLA valence coding (e.g., low/silenced -> negative, mid/unperturbed -> neutral, high/activated -> positive vs. low/silenced -> positive, mid/unperturbed neutral, high/activated -> negative)*.

*) The manuscript main findings of aIC-BLA in anxiety and valence might be synthesized and sketched into a working model in the supplementary material to better communicate the authors' working hypothesis for linking valence and anxiety in IC-amygdala circuits.

We thank the review for his comments. We have rephrased the discussion sections on aIC-BLA results to reconcile the behavior and imaging data of anxiety and valence assays (Lines 398-461), including the results of additional correlations supporting our initial interpretations (Extended data Fig. 2 and 8).

Moreover, we added [1] a summary table synthesizing our main findings, [2] a working model to represent our interpretation/discussion of somBiPOLES manipulations results, and [3] a summary of the correlation between the neural activity during anxiety and valence assay of the 3 neural populations we studied, with the level of anxiety of mice (Extended Data Fig. 8j-l).

Minor/Circuit organization

The differentiation of neurons into aIC-BLA and pIC-BLA neurons is highly appreciated. For the reader to keep with the overall flow of the manuscript, the authors may consider moving the more relevant parts of the BLA-projecting insula-resolved data (Extended Data Fig. 3g-i) to the main Fig. 4, and instead shifting the panels comparing between IC-BLA and IC-CeM projectors overall (Fig. 4k,l,m), to the Extended Data Fig. 3. This might help highlighting their key findings.

Thank you for this suggestion, we agree these interesting findings deserved to be placed in the main Figures. We have moved the results on IC-BLA projection in the main Fig. 4 and shifted the panels related to IC-BLA and IC-CeM comparison in the Extended data Fig. 4e-f. The results section has been updated accordingly (Lines 235-252).

MINOR/Additional supporting analysis

The authors infer an association of anxiety and negative valence by correlating the activity of aIC-BLA projectors in an anxiogenic space with activity post-shock (Fig. 6m). To further back-up this claim I suggest to do the equivalent analysis for positive valence with activity upon sucrose lick and move-to-port as presented in Fig. 3b,d. Additionally, a negative correlation between sucrose lick/move-to-port activity and shock responses would support the claim that the activity

upon shock in aIC-BLA projectors does indeed represent negative valence. The authors are encouraged to explore if such an analysis would lend additional support to their model.

We thank the review for this very interesting suggestion. We have thus completed the correlations already presented in Fig. 2n-o and Fig. 6m with additional correlations between neuronal activity in response to positive and negative valence.

We now provide a new Extended data Fig. 2 where we correlated aIC and pIC neuronal response to valence with anxiety-like behaviors in EPM and OFT, as well as neuronal activity to positive and negative valence (e.g activity during sucrose licks or move to the port vs. post shock).

Similarly, we provide in new Extended data Fig. 8, including correlations between [1] aIC-BLA neural activity in anxiety assays with anxiety-like behaviors, [2] aIC-BLA neural activity in valence assays with anxiety-like behaviors and [3] aIC-BLA neural activity between positive and negative valence.

Although all the correlation do not match the model, we found additional important and interesting correlations supporting our model. For example, we found that aIC-BLA neurons is negatively correlated with the time mice spent in the open arms (Fig. 6i), or that aIC-BLA neuron response to sucrose is negatively correlated to the signal during tail suspension (Extended data Fig. 8h).

Reviewer #2 (Remarks to the Author)

I have previously favorably reviewed this manuscript for Nature Neuroscience. The last version I already commented on seems identical to the one submitted here and has not been revised in light of 2 reviewers' comments (mine copied below).

As such, in light of my previous comments on this version (not shown in the rebuttal in this submission), regarding "Point #2" from my last comments, I suggest the authors consider more transparency by adding all the non-significant valence-anxiety assays correlations (shown in the rebuttal figures) to the manuscript supplemental figure. This would allow the readers to form a more comprehensive view of the relationship between valence and anxiety in aIC-BLA. After this, I recommend accepting the manuscript for publication.

We want to thank the reviewer for his positive feedback and for having "favorably reviewed it for Nature Neuroscience". We now provide all the correlations he suggested, between neuronal activity and anxiety-related behaviors, as well as between neuronal activity in response to positive vs. negative valence in Extended data Fig. 1 s-t, Extended data Fig. 2 and Extended data Fig. 8a-i.

Here is a summary of my previous comments: Following additional comments from two reviewers, in this revised version the authors have performed several additional experiments and analyses. We thank the authors for making these efforts to address reviewer concerns. Generally, our previous comments revolved around 2 main issues:

1. Establishing whether aIC, and specifically aIC-BLA, actually have a causal role in valence and anxiety.
2. Establishing a clear link between valence and anxiety in aIC-BLA.

The authors have attempted to address these issues, but we still have some concerns.

Specifically for point #1:

The authors have added 5-HT pharmacology in the aIC as a form of inhibition, which resulted in effects that are consistent with some the main claims of the paper and therefore strengthen the

manuscript. However, the results of the optogenetics experiments remain somewhat unclear. First, despite performing ex-vivo controls for BiPOLES, the in-vivo experiments yielded unclear results. The manipulation seems inconsistent with both inhibition and excitation, and the effect that is statistically significant is weak and only works for one anxiety assay but not for the other. The authors have given some arguments in the rebuttal for why only one assay was affected, but the BiPOLES effects seem more pronounced and statistically significant during the inhibition epochs and these inhibition results are inconsistent with the 5-HT pharmacological inhibition. Therefore, we feel that this BiPOLES experiment still remains inconclusive. Second, inhibition of aIC-BLA seems to have a robust effect in RTPP, but a very similar effect appears to occur with unilateral optogenetic activation of aIC-BLA, although not quite statistically significant ($p=0.07$). Why would both aIC activation and inhibition yield similar effects?

Together, these optogenetics experiments suggest that the causal role of aIC-BLA in valence and anxiety has not been really established in a robust way beyond the 5-HT pharmacology experiment. And it is unclear if this is due to methodological issues or not.

We agree with the reviewer that our study bring a lot of results to integrate and reconcile all together. Therefore, to help the readers, we provide a summary table and graphical working models (Extended data Fig. 8j-l) which we hope will help apprehend our findings. We also discuss those models (Lines 417-425).

Specifically for point #2:

The authors have indeed added correlations between some of the valence assays (foot shock, tail suspension) and anxiety assays (OFT, EPM). This is a very important and welcome addition. Nevertheless, it is unclear if the relationship between anxiety and valence in aIC and in aIC-BLA is really a robust relationship because most of the correlations are not significant, and only a few are. If this is indeed a general phenomenon, it is unclear why it is apparently inconsistent. Therefore, these experiments also give us an uneasy feeling that the link between valence and anxiety aIC-BLA has not been really established in a robust way.

We would suggest to consider more transparency by adding all the non-significant correlations (shown in the rebuttal figures) to the manuscript. This would allow the readers to form a more comprehensive view of the relationship between valence and anxiety in aIC-BLA.

We agree that providing the non-significant correlations for aIC and pIC is more transparent to the readers. Those correlations are now available in Extended Fig. 2, including a trend for a negative correlation between aIC signal during tail suspension with the time mice spent in the open arm ($p=0.102$), as well as a positive correlation for the pIC for these two variables ($p=0.049$). Additionally, we have now correlated aIC-BLA neuronal activity

[1] during anxiety assays with anxiety-like behaviors (Extended data Fig. 8a-c),

[2] during valence assays with anxiety-like behaviors (Fig. 6i and Extended data Fig. 8d-f) and

[3] between positive and negative valence (Extended data Fig. 8g-i).

REVIEWERS' COMMENTS

Reviewer #1 (Remarks to the Author):

I would like to thank the authors for their thorough revision. Their additional analyses and in particular the summary model will help to convey their key findings, as well as the open questions for future research.

With this, I am very much looking forward to seeing this published.

Reviewer #2 (Remarks to the Author):

My concerns have been addressed. I recommend accepting the manuscript for publication in Nature Communications.